# Power Load Demand Forecasting Model and Method Based on Multi-Energy Coupling

**Dunnan Liu [1,2], Lingxiang Wang [1,2,*], Guangyu Qin [1,2,*] and Mingguang Liu [1,2]**

[1]   School of Economics and Management, North China Electric Power University, Beijing 102206, China;
    50601660@ncepu.edu.cn (D.L.); examplemy163email@163.com (M.L.)
[2]   Beijing Key Laboratory of New Energy & Low Carbon Development, North China Electric Power University,
    Beijing 102206, China
*   Correspondence: 120192206112@ncepu.edu.cn (L.W.); qinguangyu@ncepu.edu.cn (G.Q.);
    Tel.: +86-1881-011-9363 (L.W.); +86-1326-346-2372 (G.Q.)

**Abstract:** At the present stage, China's energy development has the following characteristics: continuous development of new energy technology, continuous expansion of comprehensive energy system scale, and wide application of multi-energy coupling technology. Under the new situation, the accurate prediction of power load is the key to alleviate the problem that the planning and dispatching of the current power system is more complex and more demanding than the traditional power system. Therefore, firstly, this paper designs the calculation method of the power load demand of the grid under the multi-energy coupling mode, aiming at the important role of the grid in the power dispatching in the comprehensive energy system. This load calculation method for regional power grid operating load forecasting is proposed for the first time, which takes the total regional load demand and multi-energy coupling into consideration. Then, according to the participants and typical models in the multi-energy coupling mode, the key factors affecting the load in the multi-energy coupling mode are analyzed. At this stage, we fully consider the supply side resources and the demand side resources, innovatively extract the energy system structure characteristics under the condition of multi-energy coupling technology, and design a key factor index system for this mode. Finally, a least squares support vector machine optimized by the minimal redundancy maximal relevance model and the adaptive fireworks algorithm (mRMR-AFWA-LSSVM) is proposed, to carry out load forecasting for multi-energy coupling scenarios. Aiming at the complexity energy system analysis and prediction accuracy improvement of multi-energy coupling scenarios, this method applies minimal redundancy maximal relevance model to the selection of key factors in scenario analysis. It is also the first time that adaptive fireworks algorithm is applied to the optimization of adaptive fireworks algorithm, and the results show that the model optimization effect is good. In the case of A region quarterly load forecasting in southwest China, the average absolute percentage error of a least squares support vector machine optimized by the minimal redundancy maximal relevance model and the adaptive fireworks algorithm (mRMR-AFWA-LSSVM) is 2.08%, which means that this model has a high forecasting accuracy.

**Keywords:** multi-energy coupling; load forecasting; adaptive fireworks algorithm; least squares support vector machine; integrated energy system

## 1. Introduction

In China's "13th Five-Year" energy revolution, the promotion of electrification, renewable energy utilization, and distributed energy utilization has been emphasized again. The construction of comprehensive energy system is an effective way to achieve this goal. It solves the problems of

renewable energy utilization and energy efficiency by comprehensively utilizing various forms of energy and the difference between supply and demand. With the support of China's energy policy and social capital, the construction of comprehensive energy system is also continuously increasing [1,2]. However, the volatility, uneven distribution and seasonal changes of renewable energy bring great challenges to the reliable supply of renewable energy power, and directly affect the economic benefits and industrial development of the power industry [3]. Accurate load forecasting can assist power planning and decision-making, and is an important means to solve the problem of reliable power supply when multi-energy coupling sources are involved [4,5].

We need to consider its key influencing factors and model selection for load forecasting [6,7]. For example, in the scenario of charging and discharging forecasting of electric vehicles, researchers take charging facilities, users, policies, and markets as key factors and apply cloud computing, big data, artificial intelligence, and other technologies to design forecasting models [8]. Each of these works contributed to the development of the field and provided appropriate models and algorithms. However, these models and algorithms also have certain limitations. For example, any combination of prediction algorithms cannot meet the needs of multi-energy coupling scenarios. For the multi-energy coupling scenario, we also need to consider the load influencing factors and forecasting model architecture of the multi-energy coupling scenario. However, in the existing load forecasting research results, few of them are consistent with the multi-energy coupling scenario. Therefore, we need to analyze the scene characteristics and key load influencing factors according to the development status of multi-energy coupling, and then build an appropriate load forecasting model according to the analysis result and aiming at the multi-energy coupling scenario.

Comprehensive energy system is the main application scenario of multi-energy coupling technology, which has different structures and energy efficiency due to differences in energy composition, supply stability, load type, demand flexibility, and other factors [9–12]. For example, in the cold-heat-electricity system proposed by Ali Ehsan et al., they use the mutual conversion technology of cold-heat-electricity and energy storage equipment to ensure the stable supply of regional cold, hot, and electric loads. At the same time, they also realize the absorption of part of renewable energy [13]; according to the multi-energy complementary system proposed by Xuebin Wang et al., which mainly focuses on renewable energy, they reduce the volatility of thermal power generation and improve the absorption capacity of new energy through adaptive adjustment of hydropower [14]. In the demand response system proposed by Dan Wang et al., they achieved 80%–90% renewable energy absorption and 18.76% energy expenditure savings through accurate control of electric heating, interactive competition strategy of supply and demand in the power retail market, and economic means [15]. There is no doubt that these research results have made important contributions to the efficiency improvement and structural optimization of integrated energy utilization. However, they have not comprehensively analyzed the overall characteristics and energy flow characteristics of multi-energy coupling scenarios from the view of supply side and demand side, and have not pointed out the impact of multi-energy coupling on the stable supply of grid energy.

Many studies have been conducted on electric load forecasting. The construction of load forecasting model is different according to the forecast scenario and forecast content. For example, Yi Liang et al. constructed the EMD-mRMR-FOA-GRNN model (a hybrid model which combines empirical mode decomposition (EMD)), minimal redundancy maximal relevance (mRMR), general regression neural network (GRNN) with fruit fly optimization algorithm (FOA)) aiming at the nonlinearity and randomness of power load series, which is a short-term load forecasting method, and its forecasting error is [−1%, +1%] [16]; Jianzhou Wang et al. constructed a model in which the extreme learning machine (ELM), support vector machine (SVM), and least squares support vector machine (LSSVM) were used to forecast the short-term wind speed for short-term wind power forecasting, aiming at the stochastic and intermittent nature of wind power [17]. It is not difficult to find that the artificial intelligence algorithm can achieve relatively accurate load forecasting, and researchers usually improve the forecasting accuracy through trend decomposition or algorithms combination, which does



not explain the impact of its key impact factors on the load sequence from the perspective of internal relations. The summary of partial load forecasting methods is shown in Table 1.

**Table 1.** Introduction of partial load forecasting methods.

| No. | Reference | Forecast Scenario | Forecast Model | Scope of Application |
|-----|-----------|-------------------|----------------|----------------------|
| 1 | [16] | Electricity market transaction | EMD-mRMR- FOA-GRNN | Short-term load forecasting (STLF) |
| 2 | [17] | Wind power generation | ELM, SVM, and LSSVM | Short-term wind power forecast |
| 3 | [18] | Electricity market transaction | Gaussian process regression | STLF |
| 4 | [19] | Regional power planning | Box-cox transformation quantile regression and load relation factor identification | Medium and long term load forecasting and load probability density forecasting |
| 5 | [20] | — | SVM forecasting based on cointegration—granger causality test and seasonal decomposition | Monthly load forecast |
| 6 | [21] | Greenhouse gas emissions | Adaptive grey model | Annual greenhouse gas emissions |
| 7 | [22] | — | A combined method that is based on the fuzzy time series (FTS) and convolutional neural networks (CNN) | STLF |
| 8 | [23] | Generation planning and scheduling | EMD-mRMR-PSO (particle swarm optimization)-LSSVM | STLF |
| 9 | [24] | China's energy efficiency | Three-dimensional decomposition model and small-sample hybrid model | Annual energy efficiency forecast |
| 10 | [25] | Energy consumption | EEMD-ISFLA (Shuffled Frog Leaping Algorithm)-LSSVM | Annual energy consumption forecast |
| 11 | [26] | Electric power dispatching | CNN-LSTM (long short term memory) | STLF |
| 12 | [27] | Algorithm | SVM | Classification and forecasting problems |
| 13 | [28] | Power planning, operation and maintenance | MFO (Moth-Flame Optimization algorithm) -LSSVM | Annual load forecast |
| 14 | [29] | Electricity market trading and electricity distribution plan | Empirical mode decomposition, seasonal adjustment, PSO, and LSSVM model | STLF |
| 15 | [30] | Power assisted decision making | Weighted LSSVM based approach for time series forecasting | Annual load forecast |
| 16 | [31] | Demand for electricity | Deep learning framework | Seasonal and diurnal power demand forecasting |

Due to its short development history, the medium- and long-term power load demand forecasting under the multi-energy coupling scenario is characterized by a small number of samples, many influencing factors and complex interrelationships. The feature vector selection method can preliminary weaken the problems that there are many influencing factors and their interrelationships are complex [16,23]. What is more, LSSVM has excellent fitting effect for data with small samples and high latitude, and is applicable to the scenarios in this paper [27,30]. Compared with traditional forecasting methods, such as gray forecasting and time series, LSSVM has strong learning performance, and compared with emerging algorithms, such as the neural network algorithm, it has more strict data basis. In the application of LSSVM in load forecasting, Cunbin Li et al. proposed a moth flame optimization least-squares support vector machine model (MFO-LSSVM) for annual load forecasting, and its forecasting error was within ±3% [28]. Yanhua Chen et al. proposed to comprehensively optimize the least squares support vector machine by empirical decomposition, seasonal adjustment and particle swarm optimization (ESPLLSVM), and proved that this combinatorial optimization model was significantly superior to the single optimization model [29]. Current research results prove that swarm intelligence algorithm has good optimization ability for LSSVM, which can improve its forecasting accuracy. However, it is not a forecasting method aiming at multi-energy coupling scenarios, and its calculation efficiency can be improved. Adaptive fireworks algorithm (AFWA) is a new type of swarm intelligence algorithm. Its explosiveness and diversity make it highly applicable to complex scenarios such as multi-energy coupling scenario [32,33]. In this paper, we use AFWA to optimize the LSSVM model, which as the load forecasting optimization method for multi-energy coupling scenario.

For the first time, we designed a load forecasting method for multi-energy coupled scenarios, and proposed an efficient and accurate load forecasting model. Grid companies and power generation groups can benefit from this, in order to determine the optimal power planning scheme and equipment operation and maintenance scheme; At the same time, the subjects involved in the electricity market trading can use the forecast results to assist the medium- and long-term trading decisions. Specifically, this paper designs a load calculation method to meet the demand of power grid for the scenario of multi-energy coupling; The energy coupling relationship and effect under the multi-energy coupling scenario are analyzed comprehensively; and a load forecasting model of mRMR-AFWA-LSSVM for multi-energy coupling is proposed.

The first section of this paper is the introduction, discusses the development of the research content; The structure of other parts of this paper is as follows: Section 2 is the demand analysis of power load for multi-energy coupling scenario, and put forward the key influence index system of load. Section 3 briefly describes mRMR algorithm, AFWA algorithm and LSSVM algorithm, and a complete load forecasting framework is constructed. Section 4 verifies the accuracy and calculation performance of the model by combining the load-related data of a certain region in southwest China, and makes a comparative analysis and brief conclusion of the model. Section 5 makes a further discussion and summarizes the full text.

## 2. Power Load Demand Analysis

### 2.1. Calculation of Power Load Demand

In China, power grid companies are mainly responsible for the dispatch of stable and reliable electricity supply. Power load forecasting with power grid as calculation caliber is the most suitable method for load forecasting demand under the scenario of multi-energy coupling. Thus the load forecasting in this paper is aimed at the demand of power dispatching under the condition of multi-energy coupling, which uses the power grid as the calculation aperture. In the multi-energy coupling mode, partial energy self-sufficiency is realized, which makes the load demand of the region to the power grid reduce compared with the load demand without multi-energy coupling mode. Then the regional load demand should be the difference between the total power demand and the

multi-energy coupling self-sufficient energy. The calculation method of power demand based on this scenario is shown in Equation (1).

$$L_t = L_n - L_c,$$ (1)

where, $L_t$ is the load demand required by power grid dispatching, i.e., the expected result of the forecasting model in this paper; $L_t$ is the total demand of all loads; and $L_c$ is the energy self-sufficiency that is not a part of power grid dispatching in the case of multi-energy coupling.

Under the background of the continuous development of Chinese society, the total load demand will increase with the continuous growth of economy, and the self-sufficiency of multi-energy coupling energy will increase with the technological progress and social investment. Due to its advantages in new energy consumption, power expense payment, policy support and other aspects compared with the electricity provided by the grid, the society and users will give priority to the multi-energy coupling self-sufficiency to meet the load demand. The results shown in Figure 1 can be obtained by qualitative analysis of load changes.

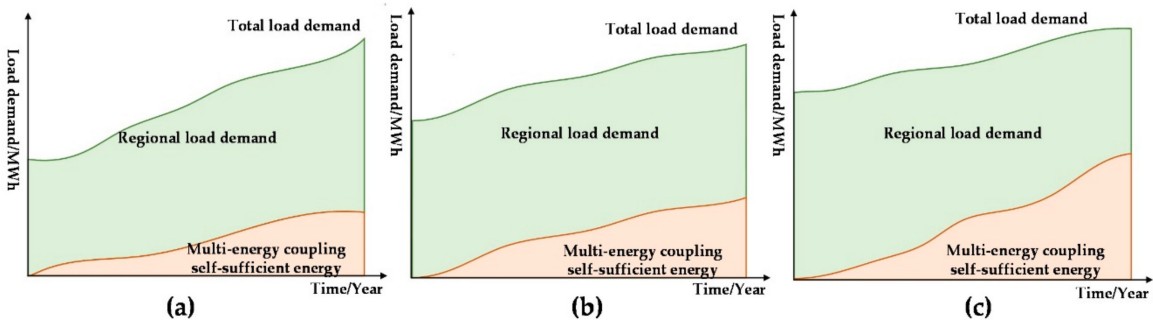

**Figure 1.** Relative changes of load demand under the scenario of multi-energy coupling. (**a**) Scenarios in which multiple coupling technologies are evolving slowly; (**b**) Scenarios in which multiple coupling technologies are developing at a moderate rate; (**c**) Scenarios in which multiple coupling technologies are evolving rapidly.

When the growth rate of total load demand is faster than that of multi-energy coupled self-sufficient power, the relative change trend of load is shown in Figure 1a. When the growth rate of multi-energy coupled self- sufficient power is faster than the growth rate of total load demand, the relative change trend is shown in Figure 1c. With the same growth rate, its relative change trend is shown in Figure 1b.

## 2.2. Multi-Energy Coupling Analysis

### 2.2.1. Subject Analysis

Load supplier generally refers to the corresponding power producer. At present, the main body of multi-energy coupling supply refers to the ways of wind, light, water, fire, storage, and other energy complementary power generation and the combined supply of cold, hot, and electricity. This is a typical way to realize the full utilization of energy resources through the collaborative supply of various energies under the background of energy Internet. It breaks the existing mode in which traditional energy supply system is independently planed and independently operated.

The main sources of load demand include commercial users, industrial users, residential users, agricultural users, government agencies, public utilities, and traffic rails. They inevitably need to use a variety of energy sources in production and life, including electricity, heat, cold, gas, coal, oil, and so on. In the context of the continuous development of integrated energy sources, multi-energy coupling sources can usually be converted to each other. Based on the different demand characteristics of different users in terms of energy type, structure proportion, time distribution, spatial distribution

and other aspects, the multi-energy coupling mode on the demand side can be designed, which is the complementary of multi-energy on the demand side.

Under the interactive coupling mode of supply and demand, the main body is still the corresponding body of supply and demand, but the boundary between supply and demand is not obvious in this process [9,10]. Participants in this process include pumped storage power station, hydrogen energy storage, virtual power plant, electric vehicle, adjustable load, etc. They reduce load and increase output during peak load period, and increase electricity consumption and reduce output during trough load period, so as to improve resource utilization efficiency and ensure energy supply.

There must be one or more of the above multi-energy coupling modes and two or more participants in a comprehensive energy system. Generally, there are multiple comprehensive energy systems in a region. Therefore, the power demand distribution structure based on the multi-energy coupling scenario is shown in Figure 2.

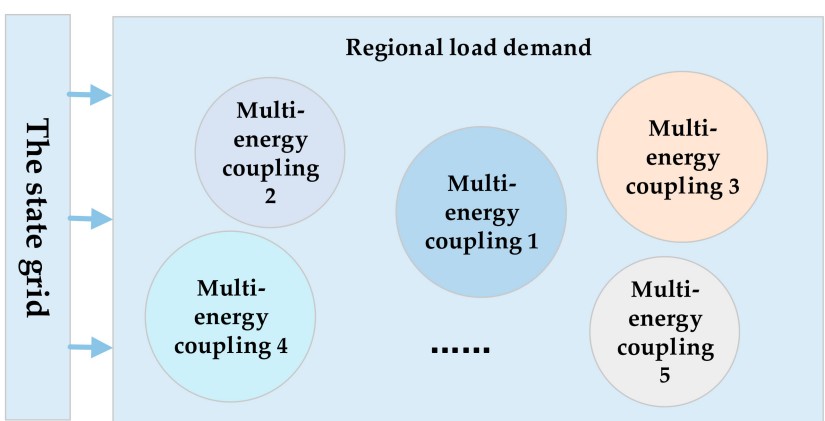

**Figure 2.** Schematic diagram of power supply and demand structure in the case of multi-energy coupling.

### 2.2.2. Multi-Energy Coupling of Typical Scenarios

With the rapid development of comprehensive energy at home and abroad, different types of multi-energy coupling utilization modes have been established in various places. This section analyzes three common multi-energy coupling utilization scenarios. Through analyzing these scenarios, we could conclude the most appropriate characteristics of the multi-energy coupling system, and analyze the key influencing factors of the multi-energy coupling energy supply.

- Coupling utilization of renewable energy

The easy access, clean and low carbon performance of renewable energy provides a foundation for its wide application in comprehensive energy systems. Wind power, hydropower, photovoltaic, distributed energy, energy storage, and other systems realize the development and utilization of renewable energy through multi-energy coupling, coordination, and complementarity.

Take the grid-connected micro-grid demonstration project of Luxi Island as an example [34]. Built on Luxi Island off the southeastern coast of China's Zhejiang province, the project is equipped with wind power of 780 kW, photovoltaic power of 300 kWp, an energy storage system of 2 MW, and residents of the island are equipped with a small single-family distributed power supply of small fans, small solar panels, and batteries. The project is built into an intelligent power supply system through wind power generation system, photovoltaic power generation system, energy storage system, single-household micro-grid system, and energy monitoring and management system. Luxi is energy self-sufficient and has the freedom to operate off-grid or off-grid. For the Luxi Island project, the integrated energy system reduces energy demand for the grid through multiple forms of energy-coupled supply.

Renewable energy sources such as wind power and photovoltaic often have large output volatility and seasonal differences. Under the influence of the geographical environment, Luxi Island has relatively stable wind power generation, which can be used as the main power supply source. At the same time, the energy storage equipment and the distributed energy at the user's house provide a guarantee for its power stability, and it can operate independently under certain conditions, in which case the power grid is not required to provide power. Therefore, the production of renewable energy needs to be fully considered before the construction of this type of multi-energy coupling system, and the resulting energy supply of the multi-energy system will eventually affect the power supply of the grid.

- Energy conservation in energy-intensive enterprises

Multi-energy coupling is often designed to achieve efficient and low-cost operation of the organization, mainly serving system functions. Therefore, many multi-energy coupling application scenarios are energy saving and consumption reduction. They improve energy efficiency through supply-side energy coupling or multi-stage energy utilization, so as to reduce energy consumption per unit product. Taking a typical industrial base as an example, the comparison of energy flow before and after the multi-energy coupling design is shown in Figure 3.

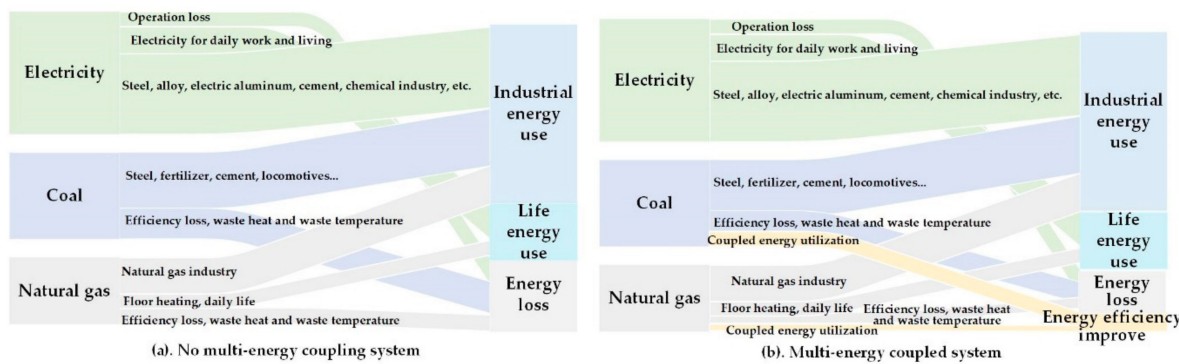

**Figure 3.** Energy flow comparison diagram of high energy consuming enterprises. (**a**) No multi-energy coupling system; (**b**) Multi-energy coupled system.

Electricity is the most efficient type of energy, with the smallest proportion of energy loss. Coal and natural gas work by converting chemical energy into heat. In this conversion process, there is energy loss caused by energy conversion efficiency, and energy loss taken away by residue and waste gas after conversion [15]. Without multi-energy coupling design, its energy flow is shown in Figure 3a. In high energy consuming enterprises with multi-energy coupling, we could improve energy efficiency by using energy saving systems to generate power with residual heat and voltage, and realize cascade utilization of energy resources. Their improvements in energy efficiency are shown in yellow in Figure 3b. Finally, the energy cost of the system is reduced, and the total load demand is reduced while the output of the product remains.

In this system, the coupling effect of multi-energy system is to improve the efficiency of energy utilization through multi-stage energy utilization and reasonable coordinated energy dispatching. The magnitude of the improvement in energy efficiency is largely related to the structural design of energy systems.

- Digital platform

In this mode, the controllable power supply and load are controlled by a digital platform. It generally includes comprehensive control platform, adjustable load, energy storage equipment, power supply equipment, and so on. Its advantage lies in the use of information control system and artificial intelligence technology to give full play to the flexibility of the system.

Take the demonstration project of alternating current charging pile in Yangzhou, Jiangsu province [35], China as an example. The region connects electric vehicle (EV) users, distributed energy, and energy storage devices through energy routers charging piles, which manage and connect external supply and load through the Internet of charging piles platform. Its structure is shown in Figure 4.

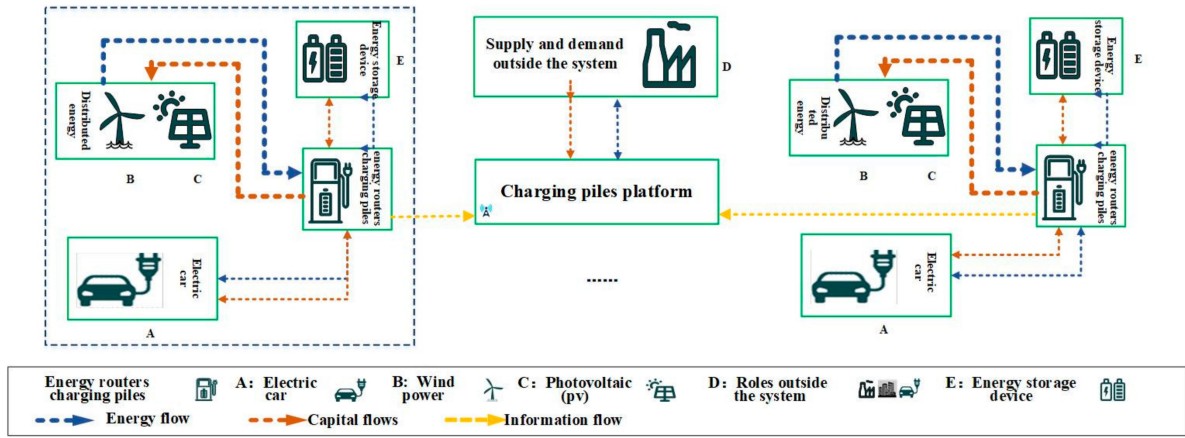

**Figure 4.** Multi-energy coupling scenario based on digital platform.

In this demonstration project, when the charging pile energy router is connected to the electric car, it can both charge and discharge, and can both fast charge and slow charge. With the permission of electric vehicle owners, electric vehicles can be used as a special energy storage device with the controlling of the digital platform. In the process of distributed energy consumption, if too much electricity is generated, the digital platform can be used to control the fast charging of electric vehicles or the energy storage to store distributed energy in a timely manner. On the contrary, the power is absorbed through integrated control.

The main body of the multi-energy system is the EV charging pile, while the charging pile is used as the media access system for other types of energy, including wind power, photovoltaic, energy storage, and distributed energy. Of cause, the scale of the connected device needs to be limited. In the process of operation, there may be imbalance between energy supply and demand, as for the instability of renewable energy. In this case, in addition to the energy storage system to meet the scheduling demand, EV charging pile operators can also achieve energy supply and demand balance through reasonable charging and discharging regulation of EV. This kind of system will have an impact on the power supply of the grid, but its fluctuation range can be controlled within a certain range. However, for the quarterly load forecasting studied in this paper, we can directly study the quarterly generation of various renewable energy sources and ignore the daily scheduling relationship between them.

### 2.2.3. Coupling Supply-Demand Mechanism Analysis

In a word, the coupling of multi-energy coupling is the integration of multi-energy resources by virtue of such characteristics as a complementary effect, substitution effect, demand flexibility, and real-time information interaction. Ultimately, we reduced energy costs and improved energy efficiency. The results of mechanism analysis in the multi-energy coupling mode are shown in Table 2.

**Table 2.** Multi-energy coupling mechanism analysis.

| | Multi-Energy on the Supply Side | Multi-Energy on the Demand Side | Interactive Coupling Mode of Supply and Demand |
|---|---|---|---|
| Subject | All kinds of renewable energy power generation manufacturers, cogeneration, energy storage, triple supply manufacturers, etc. | Commerce, industry, agriculture, government, public utilities, etc. | All kinds of energy suppliers, all kinds of energy users, Internet of vehicles, virtual power plants, etc. |
| Coupling design rationale | Spatial and temporal differences of different primary energy sources | Different loads vary in time, flexibility, and economic requirements | Spatio-temporal difference, information interaction, economy, and benefit difference between supply and demand |
| Typical coupling mode | Multi-energy coupling power generation | Multistage utilization of energy | Interaction of supply and demand |
| The coupling efficiency | Energy self-sufficiency | Energy saving | Reduce dispatch difficulty and improve economic benefit |

## 2.3. Key Influencing Factors

The factors that affect the total load demand in the area are mainly the development law of load itself and social and economic development [15,19]. For the multi-energy coupling self-sufficient power, the main influencing factors are the direct offset effect of regional energy production, the energy economic market, etc. Combined with the difficulty of data acquisition and the actual calculated data, this paper sets the key influencing factors as shown in Table 3.

**Table 3.** Key influencing factors of load demand under the scenario of multi-energy coupling.

| The Key Content | Decomposition of Content | Impact Factor |
|---|---|---|
| Total demand of all loads | Economic development | GDP current value<br>Price index |
| | Industrial structure | Contribution rate of primary industry<br>Contribution rate of secondary industry<br>Quarterly contribution rate of tertiary industry |
| | Coupling efficiency | Total fixed asset investment |
| Energy self-sufficiency | Renewable energy generation | Wind power generation<br>Solar power generation<br>Hydropower generation<br>Other renewable energy power generation |
| | Gas turbine installations, Other energy | Natural gas production<br>- |
| | Energy economy | Fuel price index<br>Investment |

In the study of various energy supplies, this paper chose the directly related power indicators, rather than the natural indicators such as climate, temperature, hydrology, and system indicators such as composition and structure. The reasons are as follows: at the present stage, the comprehensive energy system is widely distributed, the multi-energy coupling structure is not consistent, and the system size is different. It is difficult to identify the characteristics of natural conditions, and the statistical error is large, which is not conducive to forecasting. Based on the above reasons, its load forecasting calculation framework in the case of multi-energy coupling is shown in Figure 5.

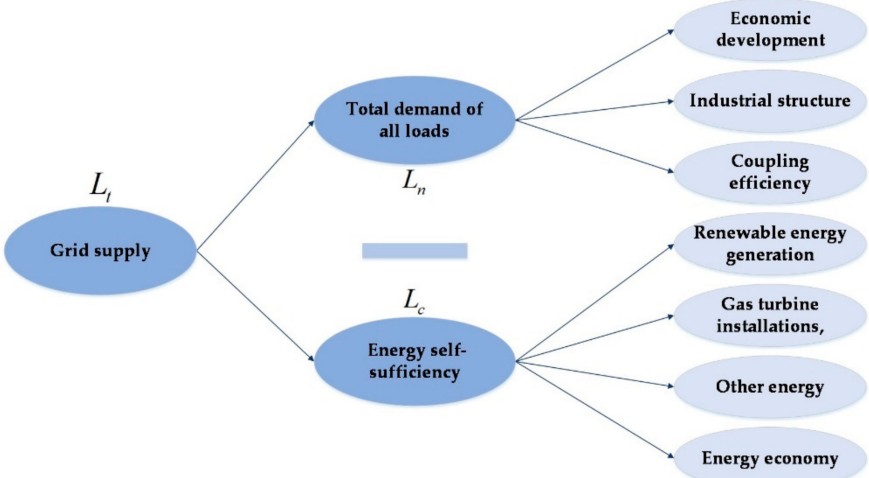

**Figure 5.** Load demand calculation with the power grid as the calculation aperture.

## 3. Model Construction

The energy demand forecasting model in the multi-energy coupling scenario is mainly composed of three parts, including the minimal redundancy maximal relevance (mRMR) model for the selection of key influencing factors, the adaptive fireworks algorithm (AFWA) algorithm for the optimization of key parameters, and the least squares support vector machine (LSSVM) model for the energy demand prediction in the multi-energy coupling model [28,36,37].

### 3.1. Feature Selection Based on Minimal Redundancy Maximal Relevance (mRMR)

Mutual information (MI) is a method to evaluate the relationship between variables. Minimal redundancy maximal relevance (mRMR) is a character selection method, which is based on MI. It maximizes the relationship between characteristic variables and target variables, and minimizes the redundant information [16]. Multi-energy coupling system often has complex system structure and energy flow relationship, and has many influencing factors. The mRMR is used to select the most important influencing factors for load forecasting under the multi-energy coupling scenario, so as to ensure the accuracy of model calculation and improve the calculation efficiency. At the same time, the basic principle of feature selection—maximum correlation information and minimum redundancy information—ensures the robustness of the model.

### 3.1.1. MI Calculation

The MI calculation could be calculated as shown in Equation (2).

$$I(x_i, y) = \iint p(x_i, y) log \frac{p(x_i, y)}{p(x_i)p(y)} dx_i dy, \tag{2}$$

where, $I(x_i, y)$ is the positive correlation between $x_i$ and $y$; $x_i$ is the characteristic variable, which presents the $i$-th influencing factor; $y$ is the target variable, which presents the load value; $p(x_i, y)$ represent the joint probability density of $x_i$ and $y$ respectively; $p(x_i)$ represent the probability density of $x_i$; and $p(y)$ represent the probability density of $y$. Through mutual information calculation, the correlation between each influencing factor and load value can be specifically measured, and the greater the correlation, the greater the value of $I(x_i, y)$.

### 3.1.2. Maximal Relevance

The maximal relevance could be calculated as shown in Equation (3).

$$maxD(S, y), D = \frac{1}{|S|} \sum_{x_i \in S} I(x_i, y), \tag{3}$$

where, $S$ is the set of influencing factors $\{x_i\}$, $|S| = m$ is the number of key influencing factors to be selected, and $D$ is the calculation result of correlation degree between m influencing factors selected and load value. Through this process, we selected the set of $m$ key influencing factors, which has the greatest correlation with the load value.

### 3.1.3. Minimum Redundancy

The minimum redundancy could be calculated as shown in Equation (4).

$$min\, R(S), R = \frac{1}{|S|^2} \sum_{x_i, x_j \in S} I(x_i, x_j), \tag{4}$$

where, $R$ is the result of redundancy calculation. Through this process, the set of m key factors with the minimum information redundancy among the influencing factors was selected.

### 3.1.4. mRMR Criteria

The mRMR criteria could be calculated as shown in Equation (5).

$$max\, \phi(D, R), \phi = D - R, \tag{5}$$

mutual information difference (MID) criterion was selected here. Based on this criterion, we used incremental search to identify the key factors, if there is already $m - 1$ feature in the hypothesis $S$, the selection basis of the $m$-th feature is shown in Equation (6).

$$\max_{x_j \in X - S_{m-1}} \left[ I(x_i, y) - \frac{1}{m-1} \sum_{x_i \in S_{m-1}} I(x_j, x_i) \right]. \tag{6}$$

### 3.2. Adaptive Fireworks Algorithm (AFWA)

Fireworks algorithm is a new search algorithm that simulates the explosion process of fireworks to conduct multi-point simultaneous explosion search [34]. Adaptive fireworks algorithm (AFWA) optimizes the algorithm by calculating the adaptive explosion range. AFWA has distributed parallelism and good adaptability [38], which is suitable for multi-energy coupling scenarios. We applied it to the optimization of kernel function width parameters and penalty parameters of LSSVM. In multi-energy scenarios, AFWA can obtain more stable and accurate calculation results compared with other algorithms [39].

### 3.2.1. FWA

The main calculation components of fireworks algorithm include explosion operator, mutation operation, mapping rule, and selection operation. For optimization problem, we could usually translate the problem into the following form:

$$\begin{aligned} min\, f(X) \\ s.t.\, g_t(X) \le 0 (t = 1, 2, \cdots, m) \end{aligned}, \tag{7}$$

where, $f(X)$ is the objective function, $g_t(X)$ is the constraint function, and $X$ is the n-dimension optimization variable. Here, the objective function was set as the minimum error between the predicted value and the actual value of load. Based on this, we interpreted the flow of AFWA algorithm.

- Initialize data

The initialization data content is Formula (8).

$$x_{ij}(0) = x_{ij}^L + rand(0,1)\left(x_{ij}^U - x_{ij}^L\right), \tag{8}$$

where, $x_{ij}(0)$ is the spatial position of the $i$-th primary fireworks in the $j$-th dimension, $x_{ij}^U$ and $x_{ij}^L$ are the upper and lower bounds of dimension respectively, and $rand(0,1)$ represents the random number generated in the direction greater than 0 and less than 1, $i = 1, 2, \cdots, N$, $j = 1, 2, \cdots, n$. In this paper, $x_i(0)$ represents the width parameters and penalty parameters of LSSVM,

- Explosion operator.

The explosion operator mainly includes the explosion intensity, explosion amplitude, and displacement variation, among which the explosion intensity is reflected as the number of sparks. The calculation method is as follows.

$$N_i = \hat{N} \bullet \frac{Y_{max} - f(X_i) + \varepsilon}{\sum\limits_{i=1}^{N} (Y_{max} - f(X_i)) + \varepsilon}, \tag{9}$$

where, $N_i$ is the number of sparks in the $i$-th fireworks, $\hat{N}$ is the constant controlling the total number of sparks, $Y_{max} = max(f(X_i))$ is the adaptive value of the individual with the worst fitness, $f(X_i)$ is the fitness value of the individual $X_i$, and $\varepsilon$ is the minimum constant preventing the denominator from being 0. Meanwhile, to prevent too many or too few sparks, we set the following rules:

$$N_i = \begin{cases} round(N_{min}), & N_i < N_{min} \\ round(N_{max}), & N_i > N_{max} \\ round(N_i), & others \end{cases}, \tag{10}$$

where, $round()$ is the integer function.

$$A_i = \hat{A} \bullet \frac{f(X_i) - Y_{min} + \varepsilon}{\sum\limits_{i=1}^{N} (f(X_i) - Y_{min}) + \varepsilon}, \tag{11}$$

where, $A_i$ is the range of explosion amplitude of the $i$-th fireworks, $\hat{A}$ is the constant limiting the maximum explosion amplitude, and $Y_{min} = min(f(X_i))$ is the adaptive value of the individuals with the best fitness.

Then, the fireworks are moved from all dimensions:

$$\Delta x_{ij} = x_{ij} + rand(0, A_i). \tag{12}$$

- Mutation operator

Here we mainly used Gaussian variation.

$$x_{ij} = x_{ij}g, \tag{13}$$

where $g \sim N(1,1)$ is the Gaussian distribution with mean and variance of 1.

- Mapping rules

The mapping rule is an algorithm that maps sparks beyond the boundary to the limited range by some method. It mainly includes the modular operation rule, specular reflection rule, random mapping rule, and so on. The modular operation rule was adopted here.

$$x_{ij} = x_{ij}^L + \left|x_{ij}^L\right|\%\left(x_{ij}^U - x_{ij}^L\right), \tag{14}$$

where, % represents modular operation, $x_{ij}$ represents the position of the *i*-th individual in the *j*-th dimension, and $x_{ij}^U, x_{ij}^L$ represents the upper and lower boundary of the *j*-th dimension respectively.

- Select operation

The selection operation in this paper adopted distance-based selection and random selection, and Euclidean distance was adopted to calculate and select the distance between two individuals.

$$R(X_i) = \sum_{q=1}^{K} d\left(X_i, X_q\right) = \sum_{q=1}^{K} \left\|X_i - X_q\right\|, \tag{15}$$

where, $R(X_i)$ represents the sum of the distance between $X_i$ and all other individuals, $d\left(X_i, X_q\right)$ represents the Euclidean distance between any two individuals $X_i, X_q$, and $K$ is the location set of all sparks after Gaussian compilation. $q \in K$.

$$p(X_i) = \frac{R(X_i)}{\sum\limits_{q \in K} R(X_i)}, \tag{16}$$

where, $p(X_i)$ represents the probability of $X_i$ being chosen. In this process, we selected the individual in the area where fireworks and sparks are most concentrated, as the optimal individuals are most likely to emerge from them.

### 3.2.2. Adaptive Adjustment

Aiming at the rationality of the calculation method of explosion radius of traditional fireworks algorithm, the adaptive fireworks algorithm uses the generated sparks to calculate the optimal fireworks explosion radius and realize the adaptive adjustment of explosion radius. The explosion radius is the distance between a specific individual and the optimal individual:

$$\hat{s} = \underset{s_i}{argmin}(d(s_i, s^*)), \tag{17}$$

$$f(s_i) > f(X), \tag{18}$$

where, $d$ is the calculation of some distance, $s_i$ is all the sparks, $s^*$ is the most adaptable individual of all the sparks and fireworks, and $X$ is the fireworks. Equations (17) and (18) show that the explosion radius is the distance satisfying the following two conditions: (1) The selection distance is the shortest distance from the candidate to the optimal individual and (2) the candidate is an individual with worse fitness than this generation of fireworks.

### 3.3. Least Squares Support Vector Machine (LSSVM)

Support vector machine (SVM) has been widely used in forecasting scenarios and has achieved many achievements [18,30]. It has excellent processing capacity for small samples and high latitude data [32], which means that it has a good applicability to the situation where the multi-energy coupling quarterly data is relatively small and the system structure is relatively complex. The least square

support vector machine (LSSVM) has a significant improvement in the prediction accuracy compared with the traditional support vector machine. Therefore, LSSVM is applied to the main load prediction algorithm under the multi-energy coupling scenario in this paper.

Nonlinear regression support vector machine (SVM) uses kernel function to change input data in sample space to high-dimensional linear eigenspace with nonlinear transformation. The linear method is used to solve nonlinear problems in the characteristic space and the global optimal solution is obtained. On the basis of standard support vector machine, the least squares support vector regression machine changes inequality constraints into equality constraints to accelerate calculation and improve accuracy.

The optimal decision function after nonlinear mapping $(\psi(x) = (\varphi(x_1), \varphi(x_2), \cdots, \varphi(x_n)))$ is as Equation (19).

$$f(x) = w^T \bullet \varphi(x) + b, \tag{19}$$

where, $w \in R^k (k > d)$ is the weight vector of high-dimensional features, which reflects the way in which each of the key impact factors affects the load and the extent to which it affects the load, $x_i \in R^d$ is the input of d-dimensional training samples, $y_i \in R$ is the output of training samples, and $b \in R$ is the bias. We searched for the optimal $w$ and $b$ based on the principle of structural risk minimization, and obtained the solution equation of the optimization problem as Equation (20).

$$min \frac{1}{2} w^T w + r \sum_{i=1}^{n} \xi_i{}^2, \tag{20}$$

where, $r > 0$ is the penalty parameter and $\xi_i$ is the relaxation variable.

Compared with the standard support vector machine algorithm, there are differences in constraint conditions.

$$y \left[ w^T \bullet \varphi(x_i) + b \right] = 1 - \xi_i, \ i = 1, 2, \cdots, n, \tag{21}$$

apply Lagrange function to solve the optimization problem, then:

$$L = \frac{1}{2} w^T \bullet w + r \bullet \frac{1}{2} \sum_{i=1}^{n} \xi_i^2 - \sum_{i=1}^{n} \alpha_i \left\{ y_i \left[ w^T \bullet \varphi(x_i) + b \right] - 1 + \xi_i \right\}$$

$$s.t. \begin{cases} w = \sum\limits_{i=1}^{n} \alpha_i y_i \varphi(x_i) \\ \sum\limits_{i=1}^{n} \alpha_i y_i = 0 \\ \alpha_i = r\xi_i \\ y_i \left[ w^T \bullet \varphi(x_i) + b \right] - 1 + \xi_i = 0 \end{cases}, \tag{22}$$

where, $\alpha_i$ is the vector of Lagrange multiplier, $\alpha_i > 0$ and $i = 1, 2, \cdots, n$. The final forecasting function can be obtained as shown in Equation (23).

$$f(x) = \sum_{i=1}^{n} \alpha_i K(x, x_i) + b, \tag{23}$$

where, $K(x_i, x_j) = \varphi(x_i)^T \varphi(x_j)$ is the kernel function satisfying Mercer condition. The kernel function in this paper is the radial basis kernel function, as shown in Formula (24).

$$K(x, x_i) = exp\left( -\frac{\|x - x_i\|^2}{2g^2} \right), \tag{24}$$

where $g$ is the width coefficient of kernel function.

*3.4. mRMR-AFWA-LSSVM Model*

We applied mRMR to select the most important feature impact factor, and then put the selected feature impact factor data content into the AFWA optimized LSSVM model for prediction. Among them, AFWA improved the model accuracy by optimizing the important parameters of support vector machine—penalty parameter and kernel function width parameter.

The main contents of this process include:

- Analyzing the possible influencing factors according to the system structure of the multi-energy scenario, collecting the corresponding data content, and conducting preliminary processing of the data. The operation here is mainly data normalization, and the dimensionless data retains the internal meaning while facilitating the calculation of the model.
- The mRMR feature factor selection process, which selects the key influencing factors at the top of the score, and they have the maximum load forecasting correlation and minimum information redundancy.
- Optimizing its width coefficient and penalty parameters by embedding AFWA into the LSSVM model.
- The training and forecasting process of LSSVM, through which the quarterly forecasting quantity of total load demand and multi-energy coupling energy supply can be obtained respectively, and the predicted value of load needed to be provided with the grid can be obtained by making the difference between them.

Finally, the energy demand forecast under the multi-energy coupling model was completed, and the forecast results could be analyzed. The model structure of mRMR-AFWA-LSSVM model for load demand forecasting is shown in Figure 6.

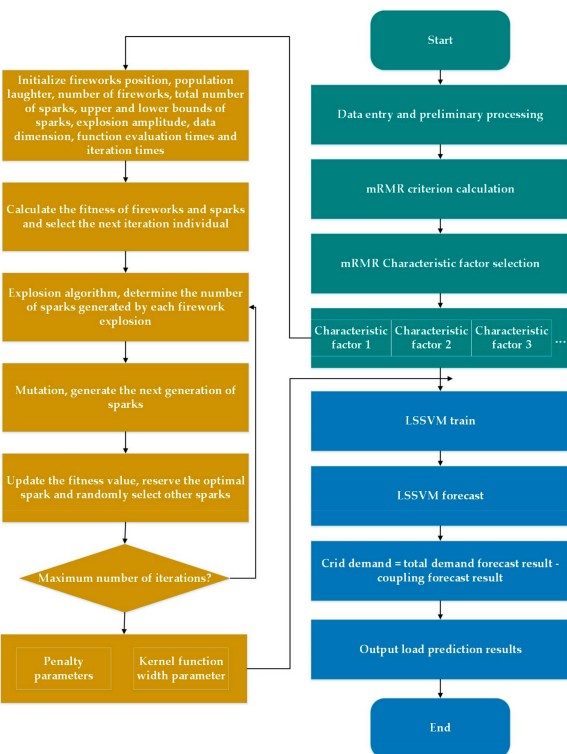

**Figure 6.** Load demand forecasting model under multi-energy coupling (least squares support vector machine optimized by the minimal redundancy maximal relevance model and the adaptive fireworks algorithm—mRMR-AFWA-LSSVM). mRMR: minimal redundancy maximal relevance; LSSVM: minimal redundancy maximal relevance.

## 4. Case Analysis

This paper made an empirical analysis of the grid load change in A region of southwest China from the first quarter of 2004 to the first quarter of 2019 (as project requirements, the data were processed). The original total load demand and multi-energy coupling supply are shown in Figure 7. It can be seen that the self-supporting amount of multi-energy coupling energy accounts for a relatively small proportion of the total quarterly load data, but it has developed rapidly in recent years, showing the characteristics of rapid development speed and drastic changes within the year. With the popularization and rapid development of multi-energy coupling, the stability of supply and demand will have higher and higher requirements on the planning and dispatching of the power grid.

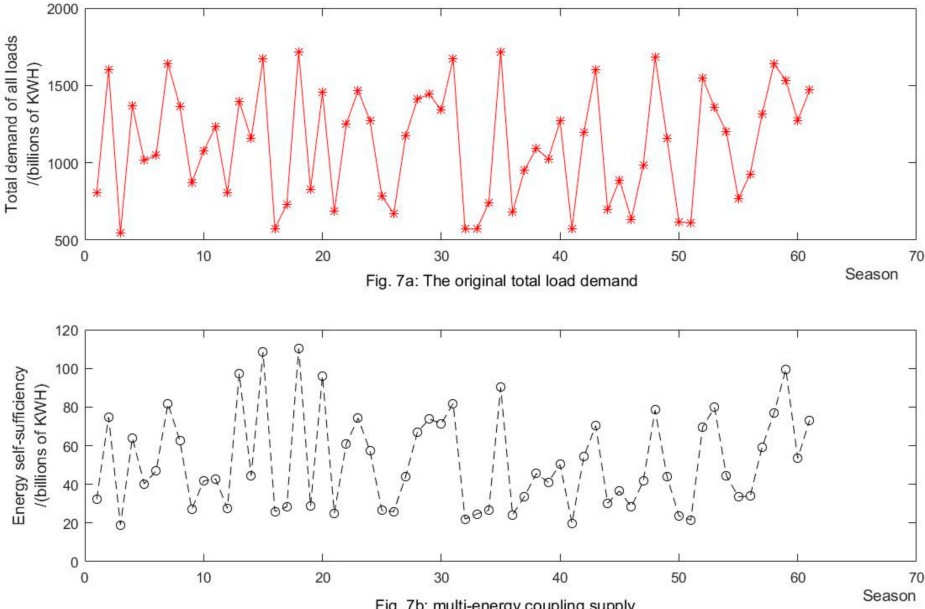

**Figure 7.** Aggregate demand data and multi-energy coupling data.(**a**) The original total load demand; (**b**) Multi-energy coupling supply.

### 4.1. mRMR Key Factor Analysis

Based on the results of multi-energy coupling scenario analysis, we obtained the corresponding load forecasting index system (as shown in Section 2.3 in this paper). First, we needed to screen the key influencing factors for load forecasting, so as to improve the efficiency of load forecasting. The incremental search method was adopted to select the feature influence factor according to Formula (6), and the selection results are shown in Tables 4 and 5. It can be seen that for the local multi-energy coupling self-load supply, the main impact is the wind and hydropower; and the main factors that affect the total load demand are price index, domestic production, and fixed self-check investment.

**Table 4.** mRMR calculation results of multi-energy coupled.

| Characteristic Factor | Solar Power Generation (10 Million KWH) | Natural Gas Production (100 Million Cubic Meters) | Wind Power Generation (Billions of KWH) | Hydropower Generation (Billions of KWH) | Other Renewable Energy Power Generation (Billions of KWH) | Investment ($10 Thousand) | Fuel Price Index |
|---|---|---|---|---|---|---|---|
| MRMR criterion | 0 | −0.16 | 1.63 | 0.94 | 0.52 | 0.62 | 0.60 |
| Rank | 6 | 7 | 1 | 2 | 5 | 3 | 4 |

**Table 5.** mRMR calculation results of total load demand.

| Characteristic Factor | Contribution Rate of Primary Industry (%) | Contribution Rate of Secondary Industry (%) | Quarterly Contribution Rate of Tertiary Industry (%) | Total Fixed Asset Investment (100 Million Yuan) | GDP Current Value ($100 Million) | Price Index |
|---|---|---|---|---|---|---|
| mRMR criterion | 0 | −0.066 | 0.73 | 0.96 | 0.94 | 1.30 |
| Rank | 5 | 6 | 4 | 2 | 3 | 1 |

The influence factors of the first three multi-energy coupling characteristics were input into the AFWA-LSSVM model for further calculation.

### 4.2. Load Forecasting With AFWA-SVM

The selected result data set of mRMR was randomly divided into test set and training set. Parameters of adaptive fireworks algorithm are set as shown in Table 6. After optimization by AFWA, the corresponding parameters are shown in Table 7.

**Table 6.** Parameter settings of adaptive fireworks algorithm.

| The Parameter Name | Values |
|---|---|
| The population size | 5 |
| The number of fireworks in gauss explosions | 5 |
| The total number of sparks | 200 |
| Spark upper bound parameter | 100 |
| Spark lower bound parameter | 0.1 |
| Blast amplitude control parameters | 100 |
| Data dimension | 2 |
| Function evaluation number | 1 |
| The number of iterations | 1 |

**Table 7.** LSSVM parameter settings after optimization.

| | Penalty Parameters (c) | Width Parameter of Kernel Function (g) |
|---|---|---|
| Multi-energy coupled forecasting | 30.9715 | 60.4294 |
| Total load forecasting | 26.4350 | 61.0462 |

After the training, we respectively used the test set and training set to make forecasting, and the predicted results are shown in Figure 8. In terms of the graph fitting effect, the model fitting degree was good and the forecasting accuracy was high. In the coupling forecasting results, the training set: MSE = 0.840196, RMSE = 0.916622, R2 = 0.998780, MAE = 0.744482, MAPE = 1.770936 and the test set: MSE = 0.661562, RMSE = 0.813365, R2 = 0.997686, MAE = 0.711322, MAPE = 2.013296. In the demand forecasting results, the training set: MSE = 2168.757653, RMSE = 46.569922, R2 = 0.984562, MAE = 37.545733, MAPE = 3.482320 and the test set: MSE = 4455.127204, RMSE = 66.746739, R2 = 0.957014, MAE = 51.794724, MAPE = 4.518415. It is not difficult to find that the forecasting accuracy was good, and the forecasting accuracy of the training set and the forecasting set was similar, thus there was no overfitting phenomenon.

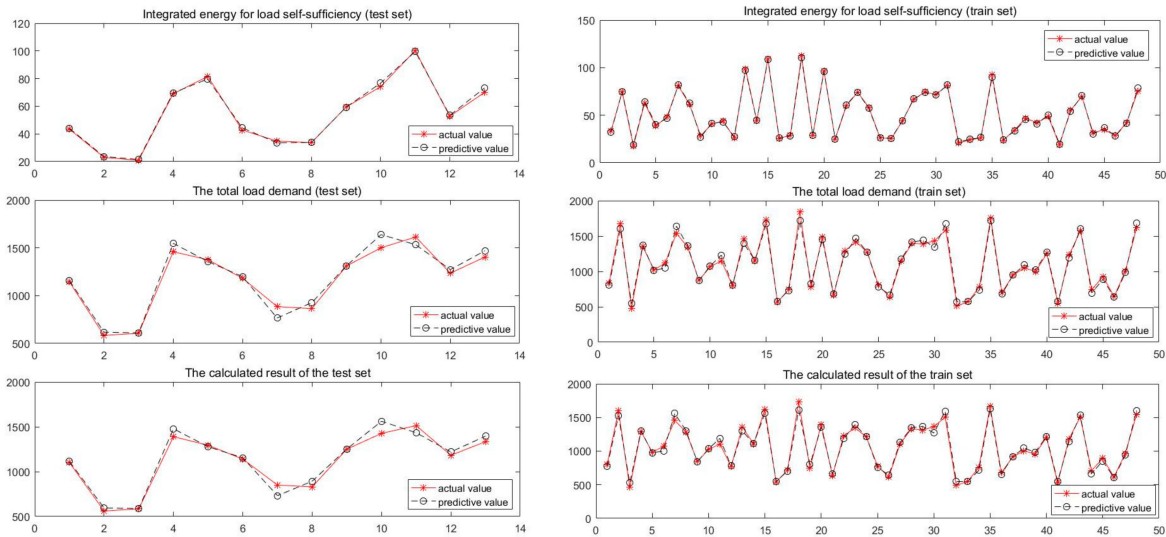

**Figure 8.** Model forecasting results.

## 4.3. Comparative Analysis and Conclusions

In order to verify the effect of the model, this paper calculated the multi-energy coupling forecasting results, total load demand forecasting results and load calculation results of mRMR-AFWA-LSSVM, AFWA-LSSVM, LSSVM, and AFWA-SVM four models respectively. The final forecasting results of the model are shown in Table 8, and in order to show the model fitting effect more intuitively, we presented the results as shown in Figure 9. Among them, mRMR-AFWA-LAAVM and AFWA-LSSVM model had the best fitting effect, LSSVM model was the second, and AFWA-SVM was the last.

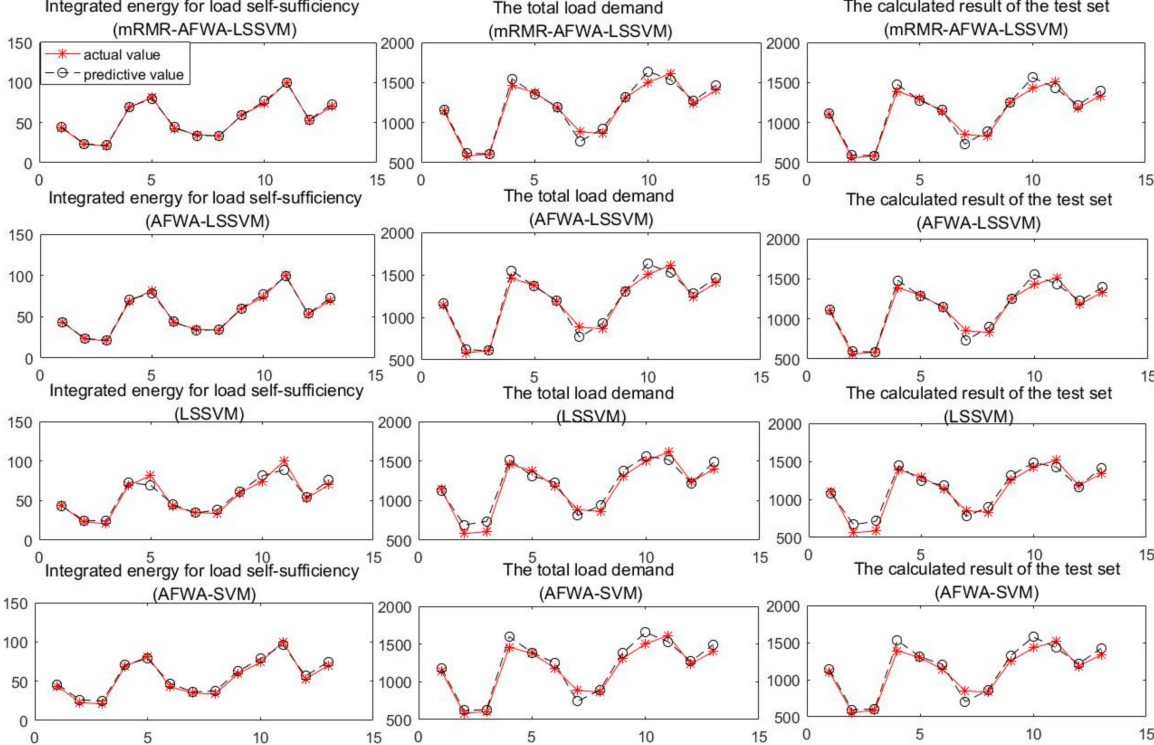

**Figure 9.** Forecasting results of four models.

**Table 8.** Forecasting results of four models.

| Actual | | | mRMR-AFWA-LAAVM | | | AFWA-LSSVM | | | LSSVM | | | AFWA-SVM | | |
|---|---|---|---|---|---|---|---|---|---|---|---|---|---|---|
| Integrated Energy | Total Load | Load | Integrated Energy | Total Load | Load | Integrated Energy | TOTAL LOAD | Load | Integrated Energy | Total Load | Load | Integrated Energy | Total Load | Load |
| 43.53 | 1144.58 | 1101.05 | 44.01 | 1157.58 | 1113.57 | 43.57 | 1160.97 | 1117.41 | 43.10 | 1121.60 | 1078.50 | 46.09 | 1183.59 | 1137.50 |
| 23.02 | 583.70 | 560.68 | 23.63 | 618.07 | 594.44 | 23.59 | 619.31 | 595.72 | 25.10 | 692.72 | 667.62 | 26.33 | 620.27 | 593.93 |
| 20.87 | 606.83 | 585.96 | 21.49 | 610.35 | 588.86 | 21.86 | 607.40 | 585.54 | 24.07 | 738.53 | 714.47 | 24.83 | 631.58 | 606.75 |
| 68.93 | 1458.72 | 1389.79 | 69.64 | 1547.29 | 1477.65 | 70.62 | 1551.04 | 1480.42 | 72.72 | 1514.04 | 1441.32 | 70.83 | 1598.52 | 1527.69 |
| 81.54 | 1376.27 | 1294.73 | 79.71 | 1357.01 | 1277.30 | 78.75 | 1363.14 | 1284.39 | 69.52 | 1311.22 | 1241.70 | 78.68 | 1386.29 | 1307.60 |
| 42.79 | 1182.28 | 1139.49 | 44.34 | 1199.14 | 1154.80 | 43.86 | 1193.60 | 1149.74 | 45.17 | 1229.65 | 1184.48 | 47.09 | 1251.71 | 1204.61 |
| 34.81 | 885.21 | 850.40 | 33.62 | 765.90 | 732.28 | 33.18 | 767.00 | 733.83 | 34.43 | 811.78 | 777.35 | 36.28 | 740.13 | 703.85 |
| 33.71 | 864.45 | 830.74 | 33.82 | 927.19 | 893.37 | 34.55 | 929.36 | 894.81 | 37.95 | 939.57 | 901.62 | 37.67 | 895.39 | 857.72 |
| 59.35 | 1310.27 | 1250.92 | 59.26 | 1312.34 | 1253.08 | 59.82 | 1306.28 | 1246.46 | 61.08 | 1378.42 | 1317.34 | 62.48 | 1383.28 | 1320.79 |
| 74.12 | 1501.11 | 1426.99 | 76.71 | 1638.69 | 1561.98 | 77.44 | 1629.99 | 1552.55 | 81.30 | 1563.00 | 1481.70 | 78.64 | 1653.84 | 1575.20 |
| 100.27 | 1613.74 | 1513.47 | 99.54 | 1533.31 | 1433.77 | 98.85 | 1528.14 | 1429.29 | 89.01 | 1515.98 | 1426.98 | 96.34 | 1530.60 | 1434.26 |
| 52.79 | 1235.36 | 1182.57 | 53.38 | 1272.74 | 1219.35 | 53.66 | 1277.24 | 1223.58 | 54.05 | 1218.30 | 1164.25 | 56.57 | 1268.14 | 1211.57 |
| 69.93 | 1404.35 | 1334.42 | 73.15 | 1469.58 | 1396.43 | 72.84 | 1464.88 | 1392.04 | 75.90 | 1492.46 | 1416.55 | 74.69 | 1495.42 | 1420.73 |

We compared and evaluated the performance of forecasting models from six aspects: model calculation time (t), mean square error (MSE), root mean square error (RMSE), average absolute error (MAE), average absolute percentage error (MAPE), determination coefficient ($R^2$), and the first five indicators are the smaller the better, but $R^2$ is the larger the better. The calculation formula is as follows.

$$MSE = \frac{1}{N}\sum_{n=1}^{N}\left(\hat{X}_{(n)} - X_{(n)}\right)^2, \tag{25}$$

$$RMSE = \sqrt{\frac{1}{N}\sum_{n=1}^{N}\left(\hat{X}_{(n)} - X_{(n)}\right)^2}, \tag{26}$$

$$MAE = \frac{1}{N}\sum_{n=1}^{N}\left|X_{(n)} - \hat{X}_{(n)}\right|, \tag{27}$$

$$MAPE = \frac{1}{N}\sum_{n=1}^{N}\left|\frac{X_{(n)} - \hat{X}_{(n)}}{X_{(n)}}\right| \times 100\%, \tag{28}$$

$$R^2 = \frac{SSR}{SST} = \frac{\sum\limits_{n=1}^{N}\left(\hat{X}_{(n)} - \overline{X}_{(n)}\right)^2}{\sum\limits_{n=1}^{N}\left(X_{(n)} - \overline{X}_{(n)}\right)^2}, \tag{29}$$

where, $N$ is the number of forecasting data groups, $n$ is the forecasting number group, $\hat{X}_{(n)}$ is the forecasting result, $X_{(n)}$ is the actual value, $\overline{X}_{(n)}$ is the actual average value, $SSR$ is the sum of squares of the regression, and $SST$ is the total sum of squares. The calculation results are as shown in Table 9.

**Table 9.** Model comparison and evaluation.

| | | Time(s) | MSE | RMSE | MAE | MAPE% | $R^2$ |
|---|---|---|---|---|---|---|---|
| Multi-energy coupling | mRMR-AFWA-LSSVM | 111.3100 | 2.050467 | 1.431945 | 1.103184 | 2.080982 | 0.996127 |
| | AFWA-LSSVM | 145.9760 | 0.661562 | 0.813365 | 0.711322 | 2.013296 | 0.997686 |
| | LSSVM | 10.5528 | 31.99167 | 5.656118 | 4.30042 | 7.468827 | 0.939572 |
| | AFWA-SVM | 10.6015 | 12.61361 | 3.551565 | 3.418807 | 7.076095 | 0.976175 |
| Total demand | mRMR-AFWA-LSSVM | 140.9960 | 4545.266 | 67.41859 | 52.3335 | 4.56392 | 0.956144 |
| | AFWA-LSSVM | 151.3300 | 4455.127 | 66.74674 | 51.79472 | 4.518415 | 0.957014 |
| | LSSVM | 8.9416 | 5877.982 | 76.668 | 70.22673 | 6.752522 | 0.943286 |
| | AFWA-SVM | 17.5940 | 7298.541 | 85.4315 | 71.40998 | 6.138677 | 0.929579 |

By analyzing the calculation results, it can be found that, for the five indexes of MSE, RMSE, MAE, MAPE, and $R^2$, the forecasting accuracy effect was AFWA-LSSVM > mRMR-AFWA-LSSVM >> LSSVM-AFWA-SVM. In the multi-energy coupling forecasting, their MAPE differs 0.07%, 5.39%, and –0.90%; and in the total demand forecasting, their MAPE differs 0.05%, 2.19%, and 0.58%. However, in the total computing time, LSSVM < AFWA-SVM << mRMR-AFWA-LSSVM < AFWA-LSSVM. The time they spent was 8.94 s, 17.5 s, 140.99 s, and 151.33 s respectively.

After mRMR selection, AFWA-LSSVM's forecasting accuracy decreased slightly and the forecasting time consumption also decreased. This is because the indicators selected in the cases used in this paper were selected according to experience, so mRMR results in a slight decrease in accuracy. However, mRMR could greatly reduce the workload of data processing in the process of actual load forecasting with large index system and abundant data in the multi-energy coupling scenario, and at the same time, it will not have a great impact on the forecasting accuracy. Therefore, mRMR-AFWA-LSSVM is still the optimal load forecasting model as a whole.

Finally, we could summarize the following model test results:

- In terms of forecasting accuracy, AFWA effectively improved the forecasting accuracy of the LSSVM model, increasing the multi-energy coupling supply forecast by about 5% and the total load forecast by about 2%. In fact, the application of LSSVM significantly improved the prediction accuracy in our experiment compared with the SVM model, which the multi-energy coupling supply forecast increased by about 6%, and the total load forecast increased by about 4%. Compared with the PSO-LSSVM model, the AFWA-LSSVM model had a certain degree of efficiency improvement, with the multi-energy coupling supply forecast increasing by about 0.4% and the total load forecast increasing by about 0.3%.

- In terms of computing efficiency, the application of mRMR significantly sped up the computing speed of the model, and could assist scene analysis to a certain extent. Those with structural connections or similar key nodes tended to have a large degree of redundancy, while those with weak correlation had a small degree of correlation.

- At the same time, aiming at the optimization problem of future application of the model, we could start from digging into the key influencing factors of demand prediction. The accuracy of multi-energy coupling forecasting was better than that of total load forecasting, its better indicator system was partly to blame.

- It could be found from the analysis that the total load demand was the most important factor affecting the power network load supply, and also the main source of errors. The appropriate influence factor selection and feature analysis would improve the accuracy of forecasting mode.

## 5. Discussion

In this paper, we proposed a least squares support vector machine optimized by the minimal redundancy maximal relevance model and the adaptive fireworks algorithm to predict the power load demand in the case of multi-energy coupling. Good results were obtained in the application of this method to the multi-energy coupling scenario in A region of southwest China. This paper could be concluded and analyzed with the following aspects:

- The power grid needs to meet the needs of power users, which will be impacted by the multi-energy coupling energy supply. The difference between the total load demand and the multi-energy coupling energy supply is the power the grid needs to provide.

- As the characteristics and energy flow of different multi-energy coupling scenarios were in great difference, we had to conclude the series of key influencing factors that affect the total load demand and multi-energy coupling supply from the energy flow of multi-energy scenarios.

- The minimal redundancy maximal relevance algorithm was applied to select the top-n critical influencing factors from the series of key influencing factors, which effectively improved the stability, accuracy, and calculation speed of the model.

- The least squares support vector machine model optimized by adaptive fireworks algorithm was used for prediction, which was more accurate than other algorithms.

In addition, the proposed least squares support vector machine optimized by the minimal redundancy maximal relevance model and the adaptive fireworks algorithm had a better prediction effect than other models in terms of calculation accuracy, but its prediction accuracy still had the possibility to be improved. As shown by the application of data in A region of southwest China, the main source of error was caused by the total load demand prediction process. Thus, an important strategy to improve the prediction accuracy of the total load demand prediction results is to build a more appropriate total load demand forecasting index system. We could take temperature, precipitation, light time, and other factors into consideration and put forward a more detailed and reliable index system, which will effectively improve the accuracy of the total load demand prediction, so as to improve the load prediction calculation results for the power grid.

Although the application of the model is subject to a variety of constraints, such as the fact that it is for multi-energy coupling scenarios, that the grid is responsible for unified dispatching, and that the

grid meets users' power needs, the applicability of the model is still very wide. The model has no limits on the size of the area, natural conditions, social conditions, and so on. This is because we added the step of scenario analysis before the prediction was carried out, which helped us list all the factors that might affect the prediction result, and select the influence factors with the maximum correlation and minimum redundancy through the minimal redundancy maximal relevance model, which ensure the robustness, applicability, and accuracy of the model. This will allow the model to perform well in both southwest and northeast China, enabling power companies to accurately predict user demand.

**Author Contributions:** Conceptualization, D.L.; methodology, L.W.; validation, D.L., L.W., G.Q. and M.L.; formal analysis, M.L.; investigation, G.Q.; writing—original draft preparation, L.W. All authors have read and agreed to the published version of the manuscript.

**Funding:** This research was supported by the National Social Science Foundation of China (No. 19ZDA081), and Technology Projects of China State Grid Corporation (SGJS0000YXJS1800187).

**Conflicts of Interest:** The authors declare no conflict of interest.

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
