# Peer review of "Power Load Demand Forecasting Model and Method Based on Multi-Energy Coupling"

_applsci, doi:10.3390/app10020584_

Round 1
Reviewer 1 Report
In the reviewer’s opinion, the paper could have been more interesting and organised better. In general, the overall contribution remains scientifically poor and technically questionable. In more detail, both the paper’s title and its Abstract state sufficiently clearly the main aims of the paper, but its novelty points remain unclear. However, the use of acronyms must be avoided in the paper’s Abstract, as they reduce its readability. Section 1 cites several references, but it does not provide an exhaustive and critical discussion of the state of the art of the related literature. Usually, the end of Section 1 should summarise the paper’s structure with the contents of its sections. Section 2 describes well established techniques. The authors should help the reader to understand the novelty aspects of the proposed methodologies. Section 3 should have provided a better description of the models under investigation, and in particular the analysis of the model affected by uncertainty, disturbance and the model-reality mismatch effects. This represents the key point when the reliability and robustness features of the proposed solutions have to be verified and validated for application to engineering systems. As already remarked for Section 2, also Section 3 in some cases contains well established topics. The authors should help the reader to follow the main points of the section. Moreover, the robustness of the proposed scheme should be analysed in a more proper way. The results described in Section 4 fail to highlight the effectiveness and the efficacy of the proposed solutions. Finally, Section 5 should have suggested more appropriate open problems and future issues that could require further investigations. On the other hand, as for the paper’s Abstract, also Section 5 should not use acronyms, since it should remain a stand-alone part of the manuscript.
Author Response
Response to Reviewer 1 Comments
Dear Editors and Reviewers:
We would like to thank applied sciences for giving us the opportunity to revise our manuscript. Thank you for your letter and the comments concerning our manuscript. Those comments are all valuable and very helpful for revising and improving our paper, as well as the important guiding significance to our researches. We have studied comments carefully and have made correction which we hope meet with approval. Revised portion are marked in red in the paper. The main corrections in the paper and the responds to the reviewer’s comments are as flowing.
Responds to the reviewer’s comments:
In more detail, both the paper’s title and its Abstract state sufficiently clearly the main aims of the paper, but its novelty points remain unclear. We agree with your suggestion and highlight the innovation of the article again in the abstract and introduction. And the analysis and model construction of the multi-energy utilization scenario in this paper is the main problem we have solved, and there are few researches worked out aiming at this scenario from this perspective, which is the most important innovation of the work. And we put forward the design of power grid load calculation method, scenario analysis strategy and key index system under the multi-energy coupling scenario, which is a targeted approach constructed from our unique perspective. The improvement of model design and prediction accuracy is another innovation of this paper. And thanks again for your suggestions.The abstract has been changed as:
“At the present stage, China's energy development has the following characteristics: continuous development of new energy technology, continuous expansion of comprehensive energy system scale, and wide application of multi-energy coupling technology. Under the new situation, the accurate prediction of power load is the key to alleviate the problem that the planning and dispatching of the current power system is more complex and more demanding than the traditional power system. Therefore, firstly, this paper designs the calculation method of the power load demand of the grid under the multi-energy coupling mode, aiming at the important role of the grid in the power dispatching in the comprehensive energy system. This load calculation method for regional power grid operating load forecasting is proposed for the first time, which takes the total regional load demand and multi-energy coupling into consideration. Then, according to the participants and typical models in the multi-energy coupling mode, the key factors affecting the load in the multi-energy coupling mode are analyzed. At this stage, we fully consider the supply side resources and the demand side resources, innovatively extract the energy system structure characteristics under the condition of multi-energy coupling technology, and design a key factor index system for this mode. Finally, a least squares support vector machine optimized by the minimal redundancy maximal relevance model and the adaptive fireworks algorithm (mRMR-AFWA-LSSVM) is proposed, to carry out load forecasting for multi-energy coupling scenarios. Aiming at the complexity energy system analysis and prediction accuracy improvement of multi-energy coupling scenarios, this method applies minimal redundancy maximal relevance model to the selection of key factors in scenario analysis. And it is also the first time that adaptive fireworks algorithm is applied to the optimization of adaptive fireworks algorithm, and the results show that the model optimization effect is good. In the case of A region quarterly load forecasting in southwest China, the average absolute percentage error of a least squares support vector machine optimized by the minimal redundancy maximal relevance model and the adaptive fireworks algorithm (mRMR-AFWA-LSSVM) is 2.08%, which means that this model has a high forecasting accuracy.”
However, the use of acronyms must be avoided in the paper’s Abstract, as they reduce its readability. Thanks for your suggestion. We have deleted the acronyms from the paper’s Abstract, the “mRMR”, “LSSVM”, ”AFWA” and “MAPE” are deleted and described in technical terms. As the acronym (mRMR-AFWA-LSSVM) is the core method proposed in this paper, the mRMR-AFWA-LSSVM is put in round brackets. Section 1 cites several references, but it does not provide an exhaustive and critical discussion of the state of the art of the related literature. Usually, the end of Section 1 should summarise the paper’s structure with the contents of its sections. Thank you for your suggestions. We have added exhaustive and critical discussions of the state of the art of the related literature in the Section 1, which are in line 54-57, line 76-81, line 87-91,and line 108-109. As for the summarize at the end of part I, we add a fuller discussion at the end of section 1. Section 2 describes well established techniques. The authors should help the reader to understand the novelty aspects of the proposed methodologies. Thanks for your comment. We have added technical description and discussion in section 2, as show in line192-194, line 200-218 and line 237-240. In terms of the innovative expression of the article, we further described in section 2, which is mainly reflected in the overview of the energy scene and the extraction of key influencing factors. In fact, we used 2.1 to introduce the innovation of this paper in dealing with power grid load supply forecasting. In 2.2, we classified the scenario of multi-energy coupling and extracted the key influencing factors of multi-energy coupling energy supply, and used 2.3 to form a comprehensive key influencing factor index system.Section 3 should have provided a better description of the models under investigation, and in particular the analysis of the model affected by uncertainty, disturbance and the model-reality mismatch effects. This represents the key point when the reliability and robustness features of the proposed solutions have to be verified and validated for application to engineering systems. Thanks for your suggestion. There is an overall revision in Section 3. We have analyzed and explained the reasons for each algorithm selection and model construction, and have given a more detailed description a of the model affected by uncertainty, disturbance and the model-reality mismatch effects.
As already remarked for Section 2, also Section 3 in some cases contains well established topics. The authors should help the reader to follow the main points of the section. Moreover, the robustness of the proposed scheme should be analyzed in a more proper way. Thanks for your suggestion. we have added some necessary notes and link-ups to help readers understand, as well as to illustrate the advantages of the corresponding algorithm. And we have reinterpreted the robustness of the model. Relevant changes can be found in line 293-297, line 302-307, line 411-417, line 447-467.
The results described in Section 4 fail to highlight the effectiveness and the efficacy of the proposed solutions. Thanks for your comment. We have added more convincing explanations and evidence in Section 4, and further discussed the effectiveness and the efficacy of the proposed model.
Finally, Section 5 should have suggested more appropriate open problems and future issues that could require further investigations. Thanks for your suggestion. we have added more appropriate open problems and future issues discussion that could be faced in the future. And there is an overall revision in Section 5.
On the other hand, as for the paper’s Abstract, also Section 5 should not use acronyms, since it should remain a stand-alone part of the manuscript. Thank you for your comments. We have made corresponding corrections in section 5, and now there are no acronyms in Section 5.

Reviewer 2 Report
The manuscript deals with an interesting technological topic of insightful remarks upon the region examined. However, the manuscript can be revised in association with the stated research objectives and, to this end, it can be published after the consideration of the following review comments suggested.
1) At the subsection 2.2.2. the renewable energy sources were treated in an integrated and holistic manner, whereas it is anticipated that no all of them should be feasibly adapted to the “Coupling utilization of renewable energy” scenario. Therefore, even though a systematic deployment has been given at section 4, a more detailed analysis of the role of renewables (advantages and disadvantages per renewable energy source to be succinctly addressed) should be mentioned here (either as projective or as real situation analysis).
2) At the subsection 2.2 there should be presented a more precise valuation/quantification of the 2.2. energy profile mentioned, in terms of values’ referred and units measured (where possible and where applicable).
3) Personally speaking, I cannot fully follow the models’ development at the four subsections 3.1 – 3.4, being bounded on solely equations’ structure. The highly bounding on mathematics is weakening the perception of the analysis. Therefore, I recommend authors to accompany the mathematic deployment with accompanied explanatory text. Then, the procedures presented at Figure 6 have to be descriptively explained at the surrounding text of subsection 3.4.
4) A distinct section “5. Discussion” has to be formulated, in which:
a) the main advantageous and disadvantageous features of the models developed, to be critically/descriptively discussed. b) A comparable evaluation Table (in narrative style per Table cell) of the key-aspects (being placed as Table rows) involved in the analysis, could be helpful. To this end, it is also recommended the valuation/narrative part of section 4 to be relocated here. c) The generalization of the research outcomes to other, than the SouthWest China, provinces of China has to conclude this “5. Discussion” section, in the light of the initially/primarily setting research objectives:
“…..Therefore, firstly, this paper designs the calculation method of the power load demand of the grid under the multi-energy coupling mode, aiming at the important role of the grid in the power dispatching in the comprehensive energy system. Then, according to the participants and typical models in the multi-energy coupling mode, the key factors affecting the load in the multi-energy coupling mode are analyzed……”.
This “5. Discussion” section can be extended up to one cross-cited extra text page. It is adequate.
5) There is a large text portion which is not cited, therefore, check and citation of all no-cited text can be considered. Meticulous check should be also taken that all Figures and Tables data to be explicitly cited at the relevant legends, where missing.
6) Due to plethora of acronyms and abbreviations it could be helpful authors to structure a nomenclature Table, in which the full names and one explanatory sentence of their functionality (along with values taken and units measured, for all variables) to be denoted.
7) Literature check and updating with more recently published and relevant to the topic papers can be undertaken. To this end, the following (indicative) list of papers can be considered/cited.
Scopus
EXPORT DATE:13 Dec 2019
Chapaloglou, S., Nesiadis, A., Iliadis, P., Atsonios, K., Nikolopoulos, N., Grammelis, P., Yiakopoulos, C., Antoniadis, I., Kakaras, E.
57195757434;55927283500;57205512308;35848147000;6507617320;6701806813;6507516045;7006017079;26643357700;
Smart energy management algorithm for load smoothing and peak shaving based on load forecasting of an island's power system
(2019) Applied Energy, 238, pp. 627-642. Cited 7 times.
https://www.scopus.com/inward/record.uri?eid=2-s2.0-85060342774&doi=10.1016%2fj.apenergy.2019.01.102&partnerID=40&md5=16867b6c3428cf88d1c287e989cc7634
DOI: 10.1016/j.apenergy.2019.01.102
AFFILIATIONS: entre for Research & Technology Hellas/Chemical Process and Energy Resources Institute, 6th km. Charilaou-Thermis, Thermi, GR 57001, Greece;
Dynamics and Structures Laboratory, Machine Design and Control Systems Section, School of Mechanical Engineering, National Technical University of Athens, Greece
ABSTRACT: In this study, a novel algorithm for the management of the power flows of an islanded power system was developed, capable of simultaneously achieving steadier conventional unit operation and shaving the demand peak values, for the days of the year that present a night peak in their load curve. The under investigation system is composed of Diesel Generators, a PV farm and a Battery Energy Storage System (BESS) with the power system's consumption to be relatively higher than its RES production. The proposed algorithm combines the use of a load forecasting methodology, a pattern recognition procedure and a custom optimal power flow scheduling algorithm. The prediction module was based on a feedforward artificial neural network, capable of short-term day ahead load forecasting. The forecasted day ahead load profile was then used as an input to the developed pattern recognition algorithm, in order to be classified based on its load curve shape (pattern). Subsequently, in case that the classification resulted in a clear night peak pattern, it was possible to estimate an hourly based trajectory for the diesel generators operation and derive the BESS charging setpoints, which result in the desired peak shaving and smoothing level simultaneously. In this way, it is possible to replace or substitute the highest power demand with stored renewable energy and to operate the diesel engines as steady as possible, diminishing the ramp up and the steep gradients before the night hours’ peak. The algorithm was integrated in the overall system model in APROS software, where dynamic simulations were performed. The simulation results proved that by applying the proposed algorithm, a combined effect of smoother diesel generator operation and peak shaving with renewable energy is achievable even with the absence of PV overproduction, diminishing the variability of the load to be covered from the conventional units. Such an operation aims at enabling diesel engines to be rated at a lower, than currently, maximum capacity while increasing the share of the renewable energy penetration into the grid. © 2019 Elsevier Ltd
AUTHOR KEYWORDS: Battery energy storage system; Energy management system; Island power systems; Load forecast; Peak shaving; Renewable energy
DOCUMENT TYPE: Article
PUBLICATION STAGE: Final
SOURCE: Scopus
Ahmad, T., Chen, H.
57198996464;55743189700;
Nonlinear autoregressive and random forest approaches to forecasting electricity load for utility energy management systems
(2019) Sustainable Cities and Society, 45, pp. 460-473. Cited 9 times.
https://www.scopus.com/inward/record.uri?eid=2-s2.0-85058677354&doi=10.1016%2fj.scs.2018.12.013&partnerID=40&md5=5f7cb959cdb98aff267cd9b435d1d13f
DOI: 10.1016/j.scs.2018.12.013
AFFILIATIONS: School of Energy and Power Engineering, Huazhong University of Science and Technology, Wuhan, Hubei, China
ABSTRACT: The capability to forecast how differences in patterns of utilization in various kinds of loads can influence energy usage is an essential effort to decrease carbon emissions and demand-side energy management. The difference in weather change starts as the first step to change the energy consumption pattern in the domestic, commercial and industrial sector. To find the change in climate and their impact on energy usage, this study examines the medium-term (MT) and long-term (LT) energy prediction for utilities, independent power producers and industrial customers to estimate the energy usage requirement of large-scale city-wide by means of using the nonlinear autoregressive model (NARM), linear model using stepwise regression (LMSR) and random forest (least square boosting) (LSBoost) approaches, based on actual environmental as well as energy consumption data. The irregular load pattern recognition to remove the abnormal trend in actual energy usage is performed by applying the outlier detection and clustering analysis. The coefficient of variation (CV) of LSBoost model is 5.019%, 3.159%, 3.292% and 3.184% in summer, autumn, winter and spring season respectively. The machine learning (ML) techniques are validated and compared based on performance and accuracy with the previously existing Gaussian process regression (GPR) model. The optimal modeling of city-wide energy demand prediction using ML-based models are guaranteed the accurate operation and design of distributed energy systems. © 2018 Elsevier Ltd
AUTHOR KEYWORDS: Energy forecasting; LMSR; LSBoost; Machine learning models; NARM
DOCUMENT TYPE: Article
PUBLICATION STAGE: Final
SOURCE: Scopus
Ahmad, T., Chen, H., Shair, J., Xu, C.
57198996464;55743189700;57193617627;57201418928;
Deployment of data-mining short and medium-term horizon cooling load forecasting models for building energy optimization and management [Déploiement de modèles prèdictifs de la charge de refroidissement à court et moyen termes pour l'exploration de données en vue de l'optimisation et de la gestion énergétiques des bâtiments]
(2019) International Journal of Refrigeration, 98, pp. 399-409. Cited 5 times.
https://www.scopus.com/inward/record.uri?eid=2-s2.0-85059028733&doi=10.1016%2fj.ijrefrig.2018.10.017&partnerID=40&md5=5bb86e06dcfaae00b9137b81a25d8a5b
DOI: 10.1016/j.ijrefrig.2018.10.017
AFFILIATIONS: School of Energy and Power Engineering, Huazhong University of Science and Technology, Wuhan, China;
Department of Electrical Engineering, Tsinghua University, Beijing, China
ABSTRACT: In this study, data-mining techniques comprising three forecasting algorithms for accurate and precise cooling load requirement prediction in the building environment, with the primary aim and the objective of improving the load management are applied. Three state-of-the-art cooling load prediction algorithms are – multiple-linear regression (MLR) model, Gaussian process regression (GPR) model and Levenberg–Marquardt backpropagation neural network (LMB-NN) model. The Pearson correlation analysis is practiced calculating the correlation between actual cooling load demand and input feature variables of climate parameters. The impact of climate variability on the building load requirement is also analyzed. Forecasting intervals are divided into two basic parts: (i) 7-day ahead prediction; and (ii) 1-month ahead prediction. To assess the prediction performance, four performance evaluation indices are applied, which are: (i) coefficient of correlation (R); (ii) mean absolute error (MAE); (iii) mean absolute percentage error (MAPE); and (iv) coefficient of variation (CV). The model's performance is compared with the selection of different hidden neurons at different load conditions. The MAPE for 7-day ahead prediction interval by MLR, GPR and LMB-NN model is 13.053%, 0.405% and 2.592%, respectively. Furthermore, the data-mining algorithms are compared and validated with the previous study, and the MAPE of Bayesian regularization neural networks is calculated 2.515% for 7-day ahead prediction. It was witnessed that the algorithms could be applied to facilitate the building cooling load prediction, by applying a relatively limited number of parameters related to energy usage as well as environmental impact in the building environment. The forecasting results show that the three algorithms are effective in predicting the irregular behavior in the data as well as cooling load demand prediction. © 2018 Elsevier Ltd and IIR
AUTHOR KEYWORDS: Building load; Cooling load prediction; Data mining models; Water source heat pump
DOCUMENT TYPE: Article
PUBLICATION STAGE: Final
SOURCE: Scopus
Li, F., Jin, G.
57211576151;57196475354;
Research on power energy load forecasting method based on KNN
(2019) International Journal of Ambient Energy, .
https://www.scopus.com/inward/record.uri?eid=2-s2.0-85074524395&doi=10.1080%2f01430750.2019.1682041&partnerID=40&md5=6748f5e3eeb3a2f810fdc77923704a7c
DOI: 10.1080/01430750.2019.1682041
AFFILIATIONS: Zhengzhou Railway Vocational & Technical College, Zhengzhou, China
ABSTRACT: To study the power energy load forecasting method using KNN algorithm. The domestic and foreign literatures are used to analyse the shortcomings of the existing power energy load forecasting methods using thge KNN algorithm and the related energy source software is used to design the power energy load forecasting system from data access, overall structure and function division through Portelt technology. After that, the application effect of functional areas such as data anomaly warning and data analysis is observed. The KNN-based power energy load forecasting method designed in this paper can accurately analyse and forecast the power load in a short period of time and has high forecasting accuracy. The results verify the validity and accuracy of the power energy load forecasting method in this paper. © 2019, © 2019 Informa UK Limited, trading as Taylor & Francis Group.
AUTHOR KEYWORDS: classification; forecasting method; KNN; liferay portal; Power energy
DOCUMENT TYPE: Article
PUBLICATION STAGE: Article in Press
SOURCE: Scopus
Xu, H., Xiong, P.
57203339291;55273936100;
Research on the power load forecasting of wave energy generation based on correlation analysis of support vector machines
(2018) Journal of Advanced Oxidation Technologies, 21 (2), art. no. 201804610, .
https://www.scopus.com/inward/record.uri?eid=2-s2.0-85051353752&doi=10.26802%2fjaots.2018.04610&partnerID=40&md5=10cf7f1bdc74f069e1798e2c45628a3e
DOI: 10.26802/jaots.2018.04610
AFFILIATIONS: School of Information Engineering, Nanchang University, Nanchang, Jiangxi, 330031, China
ABSTRACT: In this paper, the author researches on the power load forecasting of wave energy generation based on correlation analysis of support vector machines. Power load forecasting is an important part of the plan, which is based on the value of electric power load and the value of the current and the related factors. The authors discuss the environmental factors that need to be considered in different types of load forecasting based on support vector machine method, and establish the detailed process of load forecasting. The empirical analysis shows that the load forecasting method based on support vector machine has good application effect in the short term load forecasting, the forecasting error is small and the accuracy is higher. In general, the experiment result shows the proposed support vector machine algorithm can improve the overall performance in electric power load forecasting substantially. © 2018 Walter de Gruyter GmbH. All rights reserved.
AUTHOR KEYWORDS: Correlation Analysis; Power Load Forecasting; Support Vector Machines; Wave Energy Generation
DOCUMENT TYPE: Article
PUBLICATION STAGE: Final
SOURCE: Scopus
Kuo, P.-H., Huang, C.-J.
57201614237;57201619481;
A high precision artificial neural networks model for short-Term energy load forecasting
(2018) Energies, 11 (1), art. no. 213, . Cited 23 times.
https://www.scopus.com/inward/record.uri?eid=2-s2.0-85052400901&doi=10.3390%2fen11010213&partnerID=40&md5=a5ed116ce8be5929ac7c53ba6f81cb29
DOI: 10.3390/en11010213
AFFILIATIONS: Computer and Intelligent Robot Program for Bachelor Degree, National Pingtung University, Pingtung, 90004, Taiwan;
School of Electrical Engineering and Automation, Jiangxi University of Science and Technology, Ganzhou Jiangxi, 341000, China
ABSTRACT: One of the most important research topics in smart grid technology is load forecasting, because accuracy of load forecasting highly influences reliability of the smart grid systems. In the past, load forecasting was obtained by traditional analysis techniques such as time series analysis and linear regression. Since the load forecast focuses on aggregated electricity consumption patterns, researchers have recently integrated deep learning approaches with machine learning techniques. In this study, an accurate deep neural network algorithm for short-Term load forecasting (STLF) is introduced. The forecasting performance of proposed algorithm is compared with performances of five artificial intelligence algorithms that are commonly used in load forecasting. The Mean Absolute Percentage Error (MAPE) and Cumulative Variation of Root Mean Square Error (CV-RMSE) are used as accuracy evaluation indexes. The experiment results show that MAPE and CV-RMSE of proposed algorithm are 9.77% and 11.66%, respectively, displaying very high forecasting accuracy. © 2018 MDPI AG. All rights reserved.
AUTHOR KEYWORDS: artificial intelligence; Convolutional neural network; Deep neural networks; Short-Term load forecasting
DOCUMENT TYPE: Article
PUBLICATION STAGE: Final
ACCESS TYPE: Open Access
SOURCE: Scopus
Li, Y., Wen, Z., Cao, Y., Tan, Y., Sidorov, D., Panasetsky, D.
57192527885;57189067066;7404523611;36818188800;15060736400;24577257600;
A combined forecasting approach with model self-adjustment for renewable generations and energy loads in smart community
(2017) Energy, 129, pp. 216-227. Cited 18 times.
https://www.scopus.com/inward/record.uri?eid=2-s2.0-85018550165&doi=10.1016%2fj.energy.2017.04.032&partnerID=40&md5=cf48a4675621a8dac9fcb265f541cd23
DOI: 10.1016/j.energy.2017.04.032
AFFILIATIONS: College of Electrical and Information Engineering, Hunan University, Changsha, 410082, China;
State Grid Changsha Power Supply Company, Changsha, 410000, China;
Energy Systems Institute, Russian Academy of Sciences, Irkutsk, 664033, Russian Federation
ABSTRACT: The short-term forecasting of wind power, photovoltaic (PV) generation and loads is important for the secure and economical dispatching of smart community with smart grid. Considering the smart community has plenty of distributed generations, here, a concept of net load is defined as the active power difference between renewable generations (wind and PV power) and loads. Then, a combined forecasting approach, which enables to build a real-time forecasting model with parameters self-adjustment, is proposed for the forecasting of the net load in smart community. Compared with the traditional forecasting methods such as support vector machine (SVM), the proposed approach can wavily optimize the parameters of the forecasting model. Besides, an optimal method named Grid-GA searching is developed to reduce the computation time during the forecasting. Therefore, it can improve the forecasting accuracy even if there is a great of uncertainty component in wind power, PV generation and loads. Detailed case studies give a contrastive analysis of the traditional and the proposed forecasting approach. The results show that the proposed approach has the capability of self-adaption on the fluctuations of wind and PV power, and can effectively improve the forecasting accuracy and efficiency. © 2017 Elsevier Ltd
AUTHOR KEYWORDS: Combined forecasting; Photovoltaic generation; Smart community; Support vector machine; Wind power
DOCUMENT TYPE: Article
PUBLICATION STAGE: Final
SOURCE: Scopus
Su, P., Tian, X., Wang, Y., Deng, S., Zhao, J., An, Q., Wang, Y.
57195624859;56959736300;56979429400;56336107200;56154619700;25642522100;56581968500;
Recent trends in load forecasting technology for the operation optimization of distributed energy system
(2017) Energies, 10 (9), art. no. 1303, . Cited 11 times.
https://www.scopus.com/inward/record.uri?eid=2-s2.0-85029348161&doi=10.3390%2fen10091303&partnerID=40&md5=a874488ebfb35032b437b2b50d90e6ee
DOI: 10.3390/en10091303
AFFILIATIONS: Key Laboratory of Efficient Utilization of Low and Medium Grade Energy, Ministry of Education, Tianjin University, Tianjin, 300350, China
ABSTRACT: The introduction of renewable resources into the distributed energy system has challenged the operation optimization of the distributed energy system. Integration of new technologies and diversified characteristics on the demand side has exerted a great influence on the distributed energy system. In this paper, by way of literature review, first, the topological structure and the mathematical expression of the distributed energy system were summarized, and the trend of enrichment and diversification and the new characteristics of the system were evaluated. Then, the load forecasting technology was reviewed and analyzed from two aspects, fundamental research and application research. Research methods of the distributed energy system under the new trend of energies were discussed, and the boundaries of the broadened distributed energy technology were explored. © 2017 by the authors. Licensee MDPI, Basel, Switzerland.
AUTHOR KEYWORDS: Distributed energy system; Load forecasting; Renewable energy; Topological structure
DOCUMENT TYPE: Review
PUBLICATION STAGE: Final
ACCESS TYPE: Open Access
SOURCE: Scopus
Ma, W., Fang, S., Liu, G., Zhou, R.
7402703568;57193342791;55706551300;57194946216;
Modeling of district load forecasting for distributed energy system
(2017) Applied Energy, 204, pp. 181-205. Cited 34 times.
https://www.scopus.com/inward/record.uri?eid=2-s2.0-85024372380&doi=10.1016%2fj.apenergy.2017.07.009&partnerID=40&md5=cb1c9c5524a7a0d9bdb8485f3b191c1e
DOI: 10.1016/j.apenergy.2017.07.009
AFFILIATIONS: School of Energy Science and Engineering, Central South University, Changsha, 410083, China
ABSTRACT: Distributed energy system (DES) has successfully aroused increasing interests among energy policy makers and system designers, as its potential of replacing conventional energy system. The optimal modeling of district load forecasting is essential to guarantee the best design and operation of DES. This paper presents a comprehensive review of district load forecasting (DLF) models to support the application of DES. The main factors affecting district load are discussed from inside to outside, including building indoor condition, building design characteristics, district layout, local microclimate, and social & economic factors. Through classifying and comparing top-down and bottom-up methods in terms of their key features and applications, it is found that the existing methods are either lack of forecasting accuracy or burdened with forecasting workload. Previous literatures reviewed in this paper show that the hybrid forecasting models including scenario analysis, physical-statistical numerical simulation and least square support vector machine based intelligent approaches have a superior ability to balance these two contradictions under different conditions. Based on the comparison results and current trend, a framework of district load forecasting, as well as corresponding future research work, is proposed for DES planning, design and service. © 2017 Elsevier Ltd
AUTHOR KEYWORDS: Distributed energy system; Framework; Load forecasting models
DOCUMENT TYPE: Review
PUBLICATION STAGE: Final
SOURCE: Scopus
Dou, C.-X., An, X.-G., Yue, D.
7006231973;56949882000;36193477900;
Multi-agent System Based Energy Management Strategies for Microgrid by using Renewable Energy Source and Load Forecasting
(2016) Electric Power Components and Systems, 44 (18), pp. 2059-2072. Cited 7 times.
https://www.scopus.com/inward/record.uri?eid=2-s2.0-84991489121&doi=10.1080%2f15325008.2016.1210699&partnerID=40&md5=03e2a75799905fff3a6381f09e35842a
DOI: 10.1080/15325008.2016.1210699
AFFILIATIONS: Institute of Electrical Engineering, Yanshan University, Qinhuangdao, Hebei, China;
Institute of Advanced Technology, Nanjing University of Posts and Telecommunications, Nanjing, Jiangsu, China
ABSTRACT: This article focuses on multi-agent system based hierarchical energy management strategies for maximum economic and environmental benefits for microgrids. First, a two-level multi-agent based energy management system is constructed, which consists of an upper-level EMA in the view of whole system, multiple lower-level unit agents in a distributed manner, and their interactions based on communication. Second, in the upper-level agent, the energy management strategies are mainly designed by constructing multi-objective functions and by using a particle swarm optimization method based on hybrid probabilistic forecasting of renewable energy sources and loads. Third, in lower-level renewable energy source and load agents, the forecasting approach regarding renewable energy sources and loasd is mainly researched by means of the ensemble empirical mode decomposition combined with sparse Bayesian learning, called the hybrid probabilistic forecast approach. Moreover, in lower-level schedulable generation unit agents, local control strategies are also presented to regulate the output power to satisfy the reference power that is set by the upper-level agent. Finally, the validity of the proposed multi-agent based energy management strategies is demonstrated by means of simulation results. © 2016, Copyright © Taylor & Francis Group, LLC.
AUTHOR KEYWORDS: energy management strategies; ensemble empirical mode decomposition; load forecasting; microgrid; multi-agent system; multi-objective optimization; power dispatch; probabilistic forecasting; renewable energy source; sparse Bayesian learning
DOCUMENT TYPE: Article
PUBLICATION STAGE: Final
SOURCE: Scopus
Kou, P., Gao, F.
36562139800;55511023900;
A sparse heteroscedastic model for the probabilistic load forecasting in energy-intensive enterprises
(2014) International Journal of Electrical Power and Energy Systems, 55, pp. 144-154. Cited 20 times.
https://www.scopus.com/inward/record.uri?eid=2-s2.0-84884891794&doi=10.1016%2fj.ijepes.2013.09.002&partnerID=40&md5=4273f2c3133af2a5938c0b4be9248d4b
DOI: 10.1016/j.ijepes.2013.09.002
AFFILIATIONS: Systems Engineering Institute, SKLMS, Xi'An Jiaotong University, Xi'an 710049, China
ABSTRACT: The energy-intensive enterprises (EIEs) account for a significant part of the total electricity consumption in most industrial countries. In the smart grid environment, electric load forecasting in EIEs plays a critical role in the security and economical operation of both the main grid and the EIEs' micro-grid. However, the accuracy of such forecasting is highly variable due to the strong stochastic nature of the load in EIEs. In this circumstance, probabilistic forecasts are essential for quantifying the uncertainties associated with the load, thus is highly meaningful for assessing the risk of relying on the forecasts and optimizing the energy systems within EIEs. This paper focuses on the day-ahead probabilistic load forecasting in EIEs, a novel sparse heteroscedastic forecasting model based on Gaussian process is developed. With the proposed model, we can provide predictive distributions that capture the heteroscedasticity of the load in EIEs. Since the high computational complexity of Gaussian process hinder its practical application to large-scale problems such as load forecast, the proposed model employs the â„“1/2 regularizer to reduce its computational complexity, thereby enhancing its practical applicability. The simulation on real world data validates the effectiveness of the proposed model. The data used in the simulation are obtained in the real operation of an EIE in China. © 2013 Elsevier Ltd. All rights reserved.
AUTHOR KEYWORDS: Gaussian process; Heteroscedasticity; Probabilistic forecasting; Smart gird; Sparsification
DOCUMENT TYPE: Article
PUBLICATION STAGE: Final
SOURCE: Scopus
Byun, J., Hong, I., Kang, B., Park, S.
35766301000;35339051700;44861172300;8901190100;
A smart energy distribution and management system for renewable energy distribution and context-aware services based on user patterns and load forecasting
(2011) IEEE Transactions on Consumer Electronics, 57 (2), art. no. 5955177, pp. 436-444. Cited 63 times.
https://www.scopus.com/inward/record.uri?eid=2-s2.0-79960907557&doi=10.1109%2fTCE.2011.5955177&partnerID=40&md5=5af31e45d974673589931a83e52cea1c
DOI: 10.1109/TCE.2011.5955177
AFFILIATIONS: School of Electrical and Electronics Engineering, Chung-Ang University, Seoul, South Korea
ABSTRACT: Emerging green IT and smart grid technologies have changed electric power infrastructure more efficiently. These technologies enable a power system operator and a consumer to improve energy efficiency and reduce greenhouse gas emissions by optimizing energy distribution and management. There are many studies of these topics with the trend of green IT and smart grid technology. However, existing systems are still not effectively implemented in home and building because of their architectural limitations. Therefore, in this paper, we propose a smart energy distribution and management system (SEDMS) that operates through interaction between a smart energy distribution system and a smart monitoring and control system. Proposed system monitors information about power consumption, a user's situation and surroundings as well as controls appliances using dynamic patterns. Because SEDMS is connected with the existing power grid and with the newrenewable energy system, we consider integration of newrenewable energy system through electric power control. We implemented proposed system in test-bed and carry out some experiments. The results show that a reduction of the service response time and the power consumption are approximately 45.6% and 9-17% respectively. © 2011 IEEE.
AUTHOR KEYWORDS: energy distribution system; energy management system; new-renewable energy; power control; smart grid
DOCUMENT TYPE: Article
PUBLICATION STAGE: Final
SOURCE: Scopus

Author Response
Response to Reviewer 2 Comments
Dear Editors and Reviewers:
We would like to thank applied sciences for giving us the opportunity to revise our manuscript. Thank you for your letter and the comments concerning our manuscript. Those comments are all valuable and very helpful for revising and improving our paper, as well as the important guiding significance to our researches. We have studied comments carefully and have made correction which we hope meet with approval. Revised portion are marked in red in the paper. The main corrections in the paper and the responds to the reviewer’s comments are as flowing.
Responds to the reviewer’s comments:
At the subsection 2.2.2. the renewable energy sources were treated in an integrated and holistic manner, whereas it is anticipated that no all of them should be feasibly adapted to the “Coupling utilization of renewable energy” scenario. Therefore, even though a systematic deployment has been given at section 4, a more detailed analysis of the role of renewables (advantages and disadvantages per renewable energy source to be succinctly addressed) should be mentioned here (either as projective or as real situation analysis). We agree with you that " whereas it is anticipated that no all of them should be feasibly adapted to the scenario ", and we further describe the characteristics of multiple renewables in section 2.2.2, and analyze the energy composition and characteristics of each type of multi-energy coupling scenario. You can find the revision in section 2.2.2, as show in line 192-194, line 200-218 and line 237-240. In addition, this paper is a load prediction method for multi-energy coupling scenario design, so it is more inclined to take the typical coupling scenario in section 2.2 as an example to demonstrate how to extract their energy characteristics from the scenario, and then further summarize.At the subsection 2.2 there should be presented a more precise valuation/quantification of the 2.2. energy profile mentioned, in terms of values’ referred and units measured (where possible and where applicable). Thanks for your suggestion, we have added more precise instructions to the corresponding chart and text description. You can find a more precise description in 2.2, as show in line 192-194, line 200-218 and line 237-240. As some project in have not published overall data, we have to describe in an appropriate way. In fact, according to the author's idea, this part hopes to make a relatively comprehensive summary of the scene, so as to extract the key factors influencing the multi-energy coupling energy supply,so there is something missing in this part of the story.
Personally speaking, I cannot fully follow the models’ development at the four subsections 3.1 – 3.4, being bounded on solely equations’ structure. The highly bounding on mathematics is weakening the perception of the analysis. Therefore, I recommend authors to accompany the mathematic deployment with accompanied explanatory text. Then, the procedures presented at Figure 6 have to be descriptively explained at the surrounding text of subsection 3.4. Thank you for your suggestion. We have reinterpreted and explained the process of model construction, the utilization of method selection and the specific meaning of the equation. There is an overall revision in Section 3. For the specific steps in figure 6, we have made a descriptive explanation at the text of subsection 3.4.
A distinct section “5. Discussion” has to be formulated, in which:
a) the main advantageous and disadvantageous features of the models developed, to be critically/descriptively discussed. b) A comparable evaluation Table (in narrative style per Table cell) of the key-aspects (being placed as Table rows) involved in the analysis, could be helpful. To this end, it is also recommended the valuation/narrative part of section 4 to be relocated here. c) The generalization of the research outcomes to other, than the SouthWest China, provinces of China has to conclude this “5. Discussion” section, in the light of the initially/primarily setting research objectives:
“…..Therefore, firstly, this paper designs the calculation method of the power load demand of the grid under the multi-energy coupling mode, aiming at the important role of the grid in the power dispatching in the comprehensive energy system. Then, according to the participants and typical models in the multi-energy coupling mode, the key factors affecting the load in the multi-energy coupling mode are analyzed……”.
This “5. Discussion” section can be extended up to one cross-cited extra text page. It is adequate.
The main advantageous and disadvantageous features of the models developed are discussed in section 5, which can be seen in line 577-597. We have provided Comparative explanations and evaluations, but considering that part 4 needs a discussion summary and part 5 needs further explanation, we still put it in part 4.3 and changed its title to "Comparative analysis and conclusions", which can be seen in line 537-570. We have added the appropriate extended text, which can be seen in line 588-607.There is a large text portion which is not cited, therefore, check and citation of all no-cited text can be considered. Meticulous check should be also taken that all Figures and Tables data to be explicitly cited at the relevant legends, where missing. Thanks for your comments, we have added the corresponding search directory, as shown in line 179, line 200, line 231, line 247, line 278, line 297, line 302, line 336, line 339, line 412 and line 413.
Due to plethora of acronyms and abbreviations it could be helpful authors to structure a nomenclature Table, in which the full names and one explanatory sentence of their functionality (along with values taken and units measured, for all variables) to be denoted. Thanks for your comments, given that the variables mentioned in this article are not as many as they seem (they are just repeated), we've added a note for the acronyms in the corresponding section (mainly in Section 3). The variables in section 4 and their units are shown in the table header.
Literature check and updating with more recently published and relevant to the topic papers can be undertaken. To this end, the following (indicative) list of papers can be considered/cited. Thank you for your suggestion. We have added some of your suggested articles as references.

Round 2
Reviewer 1 Report
The paper has been revised according to the anonymous referee's comments and suggestions.
Reviewer 2 Report
Authors proceeded in a systematic consideration of the review comments and revised their manuscript, accordingly. The narrative flow has been smoothened and the modelling outcomes offer insightful remarks on predicting the power load demand in the case of multi-energy coupling. The proposed method has been deployed under a multi-energy coupling scenario of southwest China, where a variety of constraints unveiled the determining roles of the grid, being responsible for unified dispatching, as well as the grid meets users’ power needs. The generalization, predictability, accuracy, and applicability of the model are not narrowed to the specific Chinese provinces examined, but can enable power companies to accurately predict user demand in a global context of reference. In this respect, the manuscript can be published at the “Applied Sciences” journal as is.
